# Atmospheric deposition of reactive nitrogen to a deciduous forest in the southern Appalachian Mountains

**John T. Walker[1], Xi Chen[1,a], Zhiyong Wu[1,b], Donna Schwede[1,☆], Ryan Daly[1,c], Aleksandra Djurkovic[1],
A. Christopher Oishi[2], Eric Edgerton[3], Jesse Bash[1], Jennifer Knoepp[2,☆], Melissa Puchalski[4], John Iiames[1], and
Chelcy F. Miniat[2,d]**

[1]U.S. Environmental Protection Agency, Office of Research and Development, Durham, NC, USA

[2]U.S. Department of Agriculture, Forest Service, Southern Research Station, Coweeta Hydrologic Laboratory, Otto, NC, USA

[3]Atmospheric Research & Analysis, Inc., Cary, NC, USA

[4]U.S. Environmental Protection Agency, Office of Air and Radiation, Washington, DC, USA

[a]now at: U.S. Environmental Protection Agency, Office of Air Quality Planning and Standards, Durham, NC, USA

[b]now at: RTI International, Durham, NC, USA

[c]now at: Boulder A.I.R. LLC, Boulder, CO, USA

[d]now at: U.S. Department of Agriculture, Forest Service, Albuquerque, NM, USA

[☆]retired

**Correspondence:** John T. Walker (walker.johnt@epa.gov)

**Abstract.** Assessing nutrient critical load exceedances requires complete and accurate atmospheric deposition budgets for reactive nitrogen ($N_r$). The exceedance is the total amount of $N_r$ deposited to the ecosystem in excess of the critical load, which is the amount of $N_r$ input below which harmful effects do not occur. Total deposition includes all forms of $N_r$ (i.e., organic and inorganic) deposited to the ecosystem by wet and dry pathways. Here we present results from the Southern Appalachian Nitrogen Deposition Study (SANDS), in which a combination of measurements and field-scale modeling was used to develop a complete annual $N_r$ deposition budget for a deciduous forest at the Coweeta Hydrologic Laboratory. Wet deposition of ammonium, nitrate, nitrite, and bulk organic N were measured directly. The dry deposited $N_r$ fraction was estimated using a bidirectional resistance-based model driven with speciated measurements of $N_r$ air concentrations (e.g., ammonia, ammonium aerosol, nitric acid, nitrate aerosol, bulk organic N in aerosol, total alkyl nitrates, and total peroxy nitrates), micrometeorology, canopy structure, and biogeochemistry. Total annual deposition was $\sim 6.7\,\mathrm{kg\,N\,ha^{-1}\,yr^{-1}}$, which is on the upper end of $N_r$ critical load estimates recently developed for similar ecosystems in the nearby Great Smoky Mountains National Park. Of the total (wet + dry) budget, 51.2 % TS1 was contributed by reduced forms of $N_r$ ($NH_x$ = ammonia + ammonium), with oxidized and organic forms contributing $\sim 41.2$ % and 7.6 %, respectively. Our results indicate that reductions in $NH_x$ deposition would be needed to achieve the lowest estimates ($\sim 3.0\,\mathrm{kg\,N\,ha^{-1}\,yr^{-1}}$) of $N_r$ critical loads in southern Appalachian forests.

## 1 Introduction

Prior to the Industrial Revolution, Earth's ecosystems received reactive nitrogen ($N_r$) deposition rates of $\sim 0.5\,\mathrm{kg\,ha^{-1}\,yr^{-1}}$ (Holland et al., 1999). Since the 19th century, anthropogenic activities, both industrial and agricultural, have resulted in unprecedented quantities of $N_r$ being released into the atmosphere, subsequently altering biogeochemical cycles (Neff et al., 2002a, b; Ollinger et al., 2002; Bragazza et al., 2006; Doney et al., 2007; Galloway et al., 2008; Boonstra et al., 2017). Excessive atmospheric deposition of $N_r$ to terrestrial ecosystems may lead to soil and aquatic acidification, nutrient imbalance and enrichment,

plant damage, and microbial community changes, as well as loss of biodiversity (Bobbink et al., 1998; Lohse et al., 2008; Simkin et al., 2016). Nitrogen deposition rates in many areas, including North America, Europe, and Asia, exceed 10 kg ha$^{-1}$ yr$^{-1}$ and may double by the year 2050 in some regions (Galloway et al., 2008).

The amount of $N_r$ deposition below which significant harmful effects do not occur is known as the critical load (Nilsson and Grennfelt, 1988). Critical loads can be quantified using empirical relationships between ecosystem N input and ecosystem response (Pardo et al., 2011; Root et al., 2015), or mass-balance-type biogeochemical models (Lynch et al., 2017; McNulty et al., 2007). For the southern Appalachian Mountains, simple mass balance approaches yield critical loads similar to those derived from empirical approaches for forest health and biogeochemical responses. In a recent study employing a mass balance model for the Great Smoky Mountains National Park, Pardo et al. (2018) quantified critical loads for spruce–fir, beech, and mixed deciduous forests in the range of 2.8 to 7 kg N ha$^{-1}$ yr$^{-1}$, with the highest value corresponding to a high-elevation spruce–fir site experiencing disturbance-induced regrowth. Accurate and complete deposition budgets (i.e., including all forms of N) are required to quantify the amount of N input to ecosystems in excess of the critical load (i.e., the critical load exceedance).

Estimates of N deposition for critical load assessments can be derived from gridded chemical transport models (CTMs) (Ellis et al., 2013; Lee et al., 2016; Simkin et al., 2016; Clark et al., 2018; Makar et al., 2018), measurement–model fusion (MMF) techniques that combine measurements with CTM output (Schwede and Lear, 2014; Nanus et al., 2017; McDonnell et al., 2018; U.S. EPA, 2019), or inferential modeling with site-specific measurements (Flechard et al., 2011; Li et al., 2016). While these approaches reflect the state of the science and are widely used, they collectively suffer from incompleteness of the N deposition budget (and are therefore biased low) (Walker et al., 2019a). Monitoring networks for wet deposition (NADP/NTN) and air concentrations of $N_r$ (CASTNET) focus only on inorganic species, excluding organic forms of N, which account for $\sim 25\%$ of total N in wet deposition on average (Jickels et al., 2013). Due to difficulties in sampling (Walker et al., 2012) and the inability to fully speciate the wide range of constituents (Neff et al., 2002a; Altieri et al., 2012, 2018; Cape et al., 2011; Samy et al., 2013; Chen et al., 2018), organic N is not routinely monitored. Hence, deposition of organic N remains uncertain, and thus N deposition budgets developed from network monitoring data and CTMs remain incomplete.

Current N deposition estimates also have a relatively high degree of uncertainty in the estimation of dry deposition. While wet deposition is routinely measured, direct measurements of dry $N_r$ deposition (i.e., flux measurements) in North America are relatively few (Walker et al., 2020). Estimates of dry deposition for ecosystem assessments are therefore de-rived from models (Schwede and Lear, 2014; Li et al., 2016; Lee et al., 2016). Of the inorganic N species, $NH_3$ is the most important contributor to dry deposition in many areas (Walker et al., 2019b) but also the most uncertain (Flechard et al., 2011) due to the bidirectionality of surface–atmosphere exchange. A paucity of flux measurements (Walker et al., 2020) precludes bias correction of dry deposition in CTMs and MMF techniques, making dry deposition much more uncertain relative to wet deposition.

The Coweeta study site represents southern Appalachian Mountain forests, which are highly diverse and productive ecosystems that provide a variety of ecosystem services, including a source of surface drinking water (Caldwell et al., 2014). While deposition of oxidized N to forests in the southeastern US has declined in response to the Clean Air Act, montane ecosystems continue to receive high rates of deposition due to elevation-induced precipitation gradients (Weathers et al., 2006; Knoepp et al., 2008). Southern Appalachian forests continue to show signs of sensitivity to N deposition. For example, litterfall N fluxes and foliar N concentrations at Coweeta have steadily increased over the past 2 decades (Knoepp et al., 2018). Highly spatially variable meteorological patterns typical of complex terrain are difficult to model (Lehner and Rotach, 2018), leading to uncertainties in precipitation amounts and wet deposition (Zhang et al., 2018) as well as the micrometeorological processes that govern dry deposition (Cowan et al., 2022). Estimates of deposition from gridded CTMs in mountainous terrain therefore contain a higher degree of uncertainty relative to low-elevation ecosystems. For these reasons, a better understanding of total N deposition in southern Appalachian forests is needed.

This study investigates the N deposition budget in a remote montane forest in the southeastern US. We combine long-term (1978–2020) and seasonal intensive (2015–2016) measurements of wet deposition, speciated air concentrations of $N_r$, micrometeorology, biogeochemistry, and forest canopy structure with in situ inferential dry deposition modeling to develop an annual, speciated, total N deposition budget, including net and component $NH_3$ fluxes as well as dry and wet organic N deposition. Seasonal and annual total N deposition fluxes are presented in the context of long-term trends in air concentrations and wet deposition of inorganic N species. Spatial representativeness is characterized using measurements of air concentrations of the primary inorganic N species and previous wet deposition measurements along an elevation gradient across the topographically complex forested basin.

## 2  Methods

### 2.1  Site description

The study was conducted at the USDA Forest Service Coweeta Hydrologic Laboratory, a 2185 ha experimen-

tal forest in southwestern North Carolina, USA (35°3′ N, 83°25′ W) TS2, near the southern end of the Appalachian Mountain chain. Topography is complex, with elevations ranging from 675 TS3 to 1592 m within the Coweeta Basin.
Mean annual temperature and precipitation are 12.9 °C and 1795 mm, respectively. Dominant overstory species are *Liriodendron tulipifera*, *Quercus alba*, *Betula lenta*, and *Acer rubrum*, which comprise 24 %, 17 % TS4, 11 %, and 8 % of the basal area, respectively, in the low-elevation forests
where the study was conducted (Oishi et al., 2018). The dominant understory woody shrub species is *Rhododendron maximum* (evergreen), which comprises 13 % TS5 of the basal area (Oishi et al., 2018). Species composition in the vicinity of the eddy flux tower (EFT), further described below, is detailed in
Table S1 in the Supplement. Canopy height surrounding the EFT is ∼ 30 m.

The Coweeta Basin has been a long-term monitoring site for atmospheric chemistry and deposition since the late 1970s. Weekly wet deposition of ammonium ($NH_4^+$)
and nitrate ($NO_3^-$), along with sulfate ($SO_4^{2-}$), chloride, and base cations, has been measured as part of the NADP/NTN (Site NC25, https://nadp.slh.wisc.edu/networks/national-trends-network/, last access: 1 November 2022 TS6) since 1978. Weekly integrated air concentrations of par-
ticulate $NH_4^+$, $NO_3^-$, $SO_4^{2-}$, chloride, and base cations, as well as nitric acid ($HNO_3$) and sulfur dioxide ($SO_2$), have been measured by CASTNET (Site COW137, https://www.epa.gov/castnet, last access: 1 November 2022) since 1987. Since 2011, biweekly integrated air concentrations of am-
monia ($NH_3$) have been measured by the NADP Ammonia Monitoring Network (AMoN Site NC25, https://nadp.slh.wisc.edu/networks/ammonia-monitoring-network/, last access: 2 November 2022). Here we use these datasets to place our study results into historical context, to supplement
the more intensive atmospheric chemistry measurements described below, and to use as inputs for inferential modeling of dry deposition. The long-term NADP and CASTNET measurements are collected in the lower part of the basin, indicated as NC25/COW137 in Fig. 1.

## 2.2 Southern Appalachian Nitrogen Deposition Study

Building on the long-term NADP and CASTNET measurements described above, the Southern Appalachian Nitrogen Deposition Study (SANDS) was conducted in 2015 and 2016 to better understand the atmospheric chemistry
and deposition of reactive nitrogen at Coweeta. Intensive measurement campaigns were conducted from 21 May–9 June 2015, 6–25 August 2015, 9–26 September 2015, 19 April–11 May 2016, and 13 July–3 August 2016. A subset of measurements was conducted continuously between
February 2015 and August 2016. As described below, time-resolved and time-integrated measurement techniques were used to characterize organic N in the gas phase, in particulate matter, and in wet deposition; the temporal variability of air

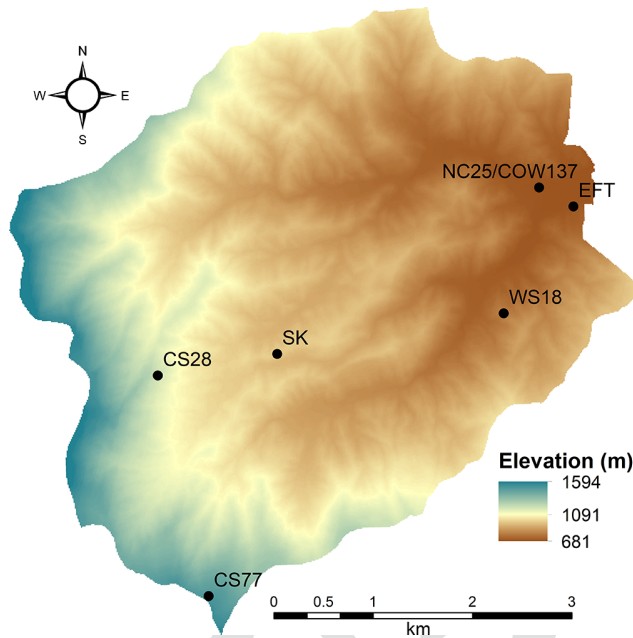

**Figure 1.** Elevation map of Coweeta Basin with sampling sites in Table 1 indicated.

concentrations of gas-phase oxidized and reduced forms of N; and the spatial variability of atmospheric N concentrations 55 across the Coweeta Basin. Vertical profiles of air concentrations were measured within the forest canopy to examine source–sink processes, and measurements of soil and vegetation chemistry were conducted to characterize $NH_3$ emission potentials. Measurements were combined with NADP 60 and CASTNET data to develop seasonal and annual total N deposition budgets employing inferential modeling for the dry deposition component. Vertical concentration profile and biogeochemical measurements were used to inform the parameterization of $NH_3$ bidirectional exchange. Sampling lo- 65 cations are described in Fig. 1 and Table 1. Measurement details are summarized in Table 2.

### 2.2.1 Wet deposition

Additional wet deposition measurements were conducted adjacent to the NTN NC25 sampler to quantify the contribution 70 of bulk water-soluble organic N (WSON) to water-soluble total nitrogen (WSTN) in precipitation. Weekly precipitation samples were collected in a modified wet-only sampler with a borosilicate glass funnel and amber glass bottle (Walker et al., 2012), shielded from sunlight, and maintained in the 75 field under continuous refrigeration to maintain the stability of ON until retrieval (Walker et al., 2012). Samples were sent to the NADP Central Analytical Laboratory on ice for analysis of $NH_4^+$, $NO_3^-$, $NO_2^-$, and WSTN as described by Walker et al. (2012). WSON concentration was calculated as 80

$$WSON = WSTN - (NH_4^+ + NO_3^- + NO_2^-). \quad (1)$$

**Table 1.** Sampling locations and type of sampler deployed.

| Site code | Latitude (N) | Longitude (W) | Elevation (m) | Sampler type |
|---|---|---|---|---|
| NC25/COW137 | 35.0605 | 83.4305 | 686 | AMoN (NC25), CASTNET (COW137), Tisch $PM_{2.5}$[c], DD-CL, TD-PC-CL, NTN (NC25), EPA precipitation |
| EFT[a] | 35.0591 | 83.4274 | 690 | MARGA[c], URG[c], passive $NH_3$ and $HNO_3$, micrometeorology |
| WS18 | 35.0512 | 83.4337 | 806 | Passive $NH_3$ and $HNO_3$ |
| SK[b] | 35.0482 | 83.4542 | 986 | Passive $NH_3$ and $HNO_3$, CASTNET (COW005) |
| CS28 | 35.0466 | 83.4650 | 1189 | Passive $NH_3$ and $HNO_3$ |
| CS77 | 35.0303 | 83.4604 | 1425 | Passive $NH_3$ and $HNO_3$ |

[a] Eddy flux tower. [b] Screwdriver Knob. [c] Tisch $PM_{2.5}$, MARGA, and URG denuder and filter pack samplers were deployed only during intensive sampling periods.

**Table 2.** Details of intensive and long-term atmospheric measurements at Coweeta.

| Sampler name | Operating periods | Measured species | Resolution | Height (m)[a] |
|---|---|---|---|---|
| DD-CL, TD-PC-CL | August 2015–August 2016 | $HNO_3$, $NO_y$, $\Sigma AN$[b], $\Sigma PN$[c] | Hourly | 8 |
| MARGA | Spring, summer 2016 intensives | $HNO_3$, $NH_3$, $NO_3^-$, $SO_4^{2-}$, $NH_4^+$ | Hourly | $\sim 40$ |
| URG denuder and filter | All intensives 2015–2016 | $HNO_3$, $NH_3$, $NO_2^-$, $NO_3^-$, $SO_4^{2-}$, $NH_4^+$ | 3 or 4 h integrated | $\sim 40$ TS7 |
| Tisch $PM_{2.5}$ | All intensives 2015–2016 | $NO_2^-$, $NO_3^-$, $SO_4^{2-}$, $NH_4^+$, WSTN | 24 h integrated | $\sim 1$ |
| CASTNET (COW137) | Long-term | $HNO_3$, $NO_3^-$, $SO_4^{2-}$, $NH_4^+$, $Cl^-$, base cations | Weekly integrated | 10 |
| AMoN (NC25) | Long-term | $NH_3$ | Biweekly passive | 10 TS8 |
| Passive $HNO_3$, $NH_3$ | 2015 | $HNO_3$, $NH_3$ | Biweekly | 10 |
| CASTNET (COW005) | 2015 | $HNO_3$, $NO_3^-$, $SO_4^{2-}$, $NH_4^+$, $Cl^-$, base cations | Weekly integrated | 10 |
| NADP/NTN | Long-term | $NO_3^-$, $NH_4^+$, $SO_4^{2-}$, $Cl^-$, $H^+$, base cations | Weekly accumulated | Ground |
| EPA precipitation | February 2015–August 2016 | $NO_2^-$, $NO_3^-$, $SO_4^{2-}$, $NH_4^+$, WSTN | Weekly accumulated | Ground |

[a] Above ground. [b] Total alkylnitrates. [c] Total peroxynitrates

The method detection limit for WSON is $10 \,\mu g \, N \, L^{-1}$ (Walker et al., 2012). These measurements were collected continuously from February 2015 to August 2016.

During the spring of 2015, thymol was added to the precipitation collection bottle as a biocide to inhibit organic nitrogen loss in the sample should the refrigerated collector malfunction or lose power. Thymol negatively affected the precision of the total nitrogen measurement, and its use was discontinued in fall of 2015. Ultimately, there were no issues with the refrigerated collector, and the thymol-containing samples were excluded from the analysis presented herein. However, this data loss resulted in a gap from August 2015 to mid-October 2015 of the 12-month period for which the total deposition budget is developed. The data gap comprised eight weekly periods in which precipitation occurred. For this period, the $NH_4^+$ and $NO_3^-$ concentrations from the collocated NADP/NTN NC25 sampler were used. Based on the SANDS measurements, the ON concentration during this period was estimated by assuming that $NH_4^+ + NO_3^-$ contributes 89 % of total nitrogen in rainfall, with WSON representing the balance (11 %). For the annual budget, weekly concentrations were combined with measured precipitation depth to calculate weekly deposition ($kg \, N \, ha^{-1}$). Comparison between SANDS and NTN concentrations of $NH_4^+$ and

$NO_3^-$ showed very good agreement (Sect. S1 in the Supplement, Fig. S1).

### 2.2.2 Air concentrations

Hourly concentrations of $NO_y$, $HNO_3$, total gas-phase peroxy nitrates ($\Sigma PN$), and total gas-phase alkyl nitrates ($\Sigma AN$) were measured continuously from August 2015 to August 2016 at the height of 8 m adjacent to the COW137 CASTNET tower (Fig. 1, Tables 1 and 2). $NO_y$ and $HNO_3$ were measured using a modified model 42S NO–$NO_2$–$NO_x$ analyzer; the $NO_y$ technique is described in detail by Williams et al. (1998). Briefly, total oxidized reactive nitrogen ($NO_y$) is converted to NO using a molybdenum catalyst heated to 325 °C. On a second channel, a metal denuder coated with potassium chloride (KCl) is used to remove $HNO_3$ before passing through a second molybdenum converter heated to 325 °C. The difference between the total $NO_y$ measurement and the $HNO_3$-scrubbed $NO_y$ measurement is interpreted as $HNO_3$. Here we refer to the method as denuder difference chemiluminescence (DD-CL).

Total peroxynitrates ($\Sigma PNs$) and total alkylnitrates ($\Sigma ANs$) were measured using a modification of the technique described by Day et al. (2002), in which PNs and ANs are thermally decomposed to $NO_2$ followed by measurement of the incremental $NO_2$ above ambient background for each decomposition step. Day et al. (2002) quantified $NO_2$ via laser-induced fluorescence, while photolytic conversion to NO and quantification of the resulting NO by NO-$O_3$ chemiluminescence is used in the current study. Here we refer to the method as thermal-decomposition photolytic-conversion chemiluminescence (TD-PC-CL). A single chemiluminescence analyzer was used for $NO_y$, $HNO_3$, $\Sigma PN$, and $\Sigma AN$ measurements. Additional detail on the instrument and associated QA/QC procedures is included in Sect. S2.

Hourly concentrations of $NH_3$ and $HNO_3$ were measured on the eddy flux tower (EFT, Fig. 1, Tables 1 and 2) at two heights above the canopy (34 and 37.5 m a.g.l., above ground, during spring 2016; 34 and 43.5 m a.g. during summer 2016) using the Monitor for Aerosols and Gases in Ambient Air (MARGA, Metrohm Applikon B.V., the Netherlands). Details and principles of the MARGA system have been previously described (Rumsey and Walker, 2016; Chen et al., 2017). Briefly, the MARGA 2S consisted of two sampler boxes positioned on the tower and a detector box located in a climate-controlled enclosure at the base of the tower. Sample boxes comprised an inlet of 1.27 cm outer diameter 30 cm long perfluoroalkoxy Teflon tubing with no particle size selection, through which airflow was mass-controlled at ~ 16.7 L min$^{-1}$, a wet rotating denuder (WRD) for collection of soluble gases, and a steam jet aerosol collector (SJAC). Liquid sample from the WRD and SJAC is continuously drawn from the sample boxes down the tower to the analytical box for analysis by ion chromatography (IC) on an hourly basis at the detector unit located in a climate-controlled en-

closure at the base of the tower. At the beginning and end of the measurement intensive, multi-level liquid $NO_3^-$ and $NH_4^+$ standards were introduced at the WRD and SJAC, with airflow turned off, to assess the analytical accuracy of the $NH_3$ and $HNO_3$ measurement. MARGA measurements were conducted during the spring and summer 2016 intensives. Comparisons of continuous and time-integrated methods for $HNO_3$ and $NH_3$ are summarized in Sect. S1 (Figs. S2 and S3).

Concentrations of $NH_3$, $HNO_3$, $SO_2$, $NH_4^+$, $NO_3^-$, and $SO_4^{2-}$ in air were measured concurrently on the EFT at 10 heights from just above the forest floor (0.5 m a.g.) to several meters above the canopy (upper height of 37 m a.g. during spring 2016, 43.5 m a.g. during summer 2016) using a glass annular denuder and filter pack (URG Corporation, Chapel Hill, NC) system. The sampling assembly included a 1 % $Na_2CO_3$-coated denuder for collection of acid gases followed by a 1 % $H_3PO_3$-coated denuder for collection of $NH_3$, a filter pack containing a primary Teflon filter for collection of aerosol, and a backup Nylon filter (47 mm, Pall Corp, Port Washington, NY) to collect $HNO_3$ liberated by dissociation of $NH_4NO_3$ on the primary filter. Inlets were Teflon-coated glass impactors with a nominal 2.5 μm aerodynamic diameter cut point (URG Corporation, Chapel Hill, NC). Sample durations were typically 3 or 4 h at a flow rate of ~ 16.7 L min$^{-1}$. Flow rates were controlled by critical orifice and were verified before and after each sampling period with a NIST-traceable primary standard flowmeter (Bios DryCal DC-Lite flowmeter, Mesa Laboratories, Inc., Lakewood, CO).

Denuders and filters were extracted with 10 mL of deionized water and analyzed by ion chromatography (IC, Dionex model ICS-2100, Thermo Scientific, Waltham, MA). Extracts were analyzed for cations using Dionex IonPac 2 mm CG12 guard and CS12 analytical columns; separations were conducted using 20 mM methanesulfonic acid (MSA) as eluent at a flow rate of 0.25 mL min$^{-1}$. Anions were analyzed (IonPac 2 mm AG23 guard column, AS23 analytical columns) using an isocratic eluent mix of carbonate–bicarbonate (4.5 : 0.8 mM) at a flow rate of 0.25 mL min$^{-1}$. Multi-point ($\geq 5$) calibrations were conducted using a mixture prepared from individual inorganic standards (Inorganic Ventures, Christiansburg, VA). A midlevel accuracy check standard was prepared from certified standards mix (AccuStandard, New Haven, CT) for quality assurance and quality control. Profile measurements were conducted during each of the five SANDS intensives.

Bulk organic nitrogen in aerosols was measured using a high-volume (Hi-Vol) Tisch TE-1000 (Tisch Environmental, Cleves, OH) dual cyclone $PM_{2.5}$ sampler operated at a flow rate of 230 L min$^{-1}$. The unit was deployed at ground level adjacent to the COW137 CASTNET tower and collected 24 h (started at 07:00 local time) integrated samples on pre-baked (550 °C for 12 h) quartz-fiber (QF) filters (90 mm, Pall Corp, Port Washington, NY). Field blanks were collected the same

way except being loaded in the sampler without the pump switched on. A QF punch (1.5 cm$^2$) from each sample was extracted with deionized water (18.2 MΩ cm, Milli-Q reference system, Millipore, Burlington, MA) in an ultrasonic bath for 45 min. The sample extract was filtered through a 0.2 μm pore size polytetrafluoroethylene membrane syringe filter (Iso-disc, Sigma Aldrich, St. Louis, MO) before subsequent analyses.

Water-soluble total N (WSTN) concentrations were measured using a high-temperature catalytic combustion and chemiluminescence method that included a total organic carbon analyzer (TOC-VCSH) combined with a total nitrogen module (TNM-1) (Shimadzu Scientific Instruments, Columbia, MD). Briefly, the TN module converts all nitrogen compounds to NO at 720 °C in a combustion chamber, and NO is quantified by NO$_2$ chemiluminescence through reaction with ozone. A five-point calibration was conducted with KNO$_3$ standard solution for each batch of samples. Before and after each batch of samples was analyzed, quality assurance checks and analyses, including lab deionized and accuracy check standards, were conducted to ensure accuracy and precision. Inorganic species (NH$_4^+$, NO$_3^-$, NO$_2^-$) were analyzed by IC as described above, and WSON was calculated according to Eq. (1). Comparisons of Hi-Vol and CASTNET PM measurements are summarized in Sect. S1 (Fig. S4).

To evaluate the spatial distribution of gaseous N across the Coweeta Basin, additional passive sampling of HNO$_3$ and NH$_3$ was conducted across an elevation gradient for the full year of 2015 (Fig. 1, Tables 1 and 2). Samplers were deployed for 2-week periods at the height of 10 m a.g. on an aluminum tilt-down tower. NH$_3$ measurements followed AMoN methods. HNO$_3$ was collected on 47 mm Nylon filters (Nylasorb, Pall Corp, Port Washington, NY) as described by Bytnerowicz et al. (2005). NH$_3$ and HNO$_3$ sampler preparation and analysis was performed by the NADP Central Analytical Laboratory and CASTNET laboratories, respectively. Field calibration of the passive HNO$_3$ measurements (Fig. S5) was based on comparison with a collocated CASTNET sampler at the Screwdriver Knob site (Fig. 1, Tables 1 and 2), which also operated for the full year of 2015.

### 2.2.3 Micrometeorology

Site characteristics of micrometeorology and ecosystem fluxes of water and carbon dioxide have been previously described (Novick et al., 2013, 2014; Oishi et al., 2018). Three-dimensional wind components were measured by a sonic anemometer (model 81000, R.M. Young Company, Traverse City, MI) above the forest canopy on the EFT. Momentum and kinematic heat fluxes were determined by eddy covariance (EC) from sonic data. For EC calculations, raw 10 Hz sonic data were processed into hourly averages after block average detrending and 2D coordinate rotation (Novick et al., 2013). Air temperature and relative humidity (RH) were measured at the top of the tower (EC155, Camp-

bell Scientific, Logan, UT) and 2/3 canopy height (HMP-45, Vaisala, Helsinki, Finland). Photosynthetically active radiation (PAR; LI-190, LI-COR Biosciences, Lincoln, NE) as well as upward and downward shortwave and longwave radiation (CNR 4, Kipp & Zonen, Delft, the Netherlands) were measured at the top of the tower. Surface wetness was measured in the canopy crown, in the understory, and at the ground using leaf wetness sensors (model 237, Campbell Scientific). Soil volumetric water content (VWC) averaged over 0–30 cm depth was measured in four locations around the tower using water content reflectometers (CS616, Campbell Scientific). Soil temperature was measured at four depths (5, 20, 35, and 55 cm) in two locations near the tower using thermistors. For missing data, linear interpolation was used to fill short gaps (1–4 h). Longer gaps were filled by substitution using the average hourly diel profile calculated for each month. Micrometeorological data were used for inferential dry deposition modeling as described below.

### 2.2.4 Biogeochemistry

Ammonia emission potentials (Γ) and compensation points for live vegetation, leaf litter on the forest floor, and soil were estimated from measurements of NH$_4^+$ and pH in the leaf and litter tissue and soil pore water (Massad et al., 2010). Green leaves were collected from 18 species (Table S1) within the flux footprint of the tower and other locations in the Coweeta Basin. Leaf litter was collected along transects extending ∼ 100 m to the northeast and southwest (i.e., predominant wind directions) of the flux tower. Litter included a composite of intact leaves and leaf fragments and excluded the more decomposed material at the top of the organic soil layer. Approximately 5 g of leaf tissue was ground in liquid nitrogen using a mortar and pestle and small coffee grinder, then extracted with 20 mL of deionized water. pH was determined directly on the extracts (Oakton pH 2100 m, Mettler Toledo InLab Micro electrode). The [NH$_4^+$] in the extracts, which reflects the bulk tissue concentration, was determined by ion chromatography as described above for denuder measurements either directly or, for samples with high organic content, after separation of the NH$_4^+$ from the solution as NH$_3$ using headspace equilibration. For the headspace method, 5 mL of tissue extract was added to a 250 mL high-density polyethylene jar containing two ALPHA passive samplers (Center for Ecology and Hydrology; Tang et al., 2001), without the diffusion barrier, affixed to the interior of the lid. The jar was sealed, and 5 mL of 0.3 N NaOH was added to the extract via a septum. The NH$_3$ liberated from the liquid extract into the headspace was collected by the passive diffusion samplers over a period of 48 h, after which the passive sampler was extracted with 10 mL of deionized water. Extracts were then analyzed by ion chromatography as described above.

Emission potentials of the vegetation (Γ$_s$) and litter (Γ$_l$) were estimated from measured concentrations of [H$^+$] (M)

and $[NH_4^+]$ ($\mu g\,g^{-1}$ tissue fresh weight) in the bulk tissue as

$$\Gamma_{s,1} = \frac{[NH_4^+] \times (5.56 \times 10^{-5}) \times LD}{[H^+]}, \tag{2}$$

where LD is leaf density ($kg\,L^{-1}$ fresh tissue, equivalent to $g\,cm^{-3}$ fresh tissue). In this case, LD values for woody deciduous and woody evergreen species are 0.37 and 0.42 $kg\,L^{-1}$, respectively (Poorter et al., 2009). Emission potentials for litter, which consisted of a mix of intact or partial leaves and needles, assume an average density of 0.4 $kg\,L^{-1}$. The factor of $5.56 \times 10^{-5}$ in Eq. (2) is necessary to convert $[NH_4^+]$ from $\mu g\,NH_4^+\,g^{-1}$ tissue to $mol\,NH_4^+\,kg^{-1}$ tissue.

Soil chemistry was measured in 20 m × 20 m plots located in the vicinity of the tower. During 2010, soil $NH_4^+$ was determined on soil samples collected with polyvinyl chloride cores (5 cm diameter and 10 cm deep) in four locations (replicates) within each of four plots. Samples were collected bimonthly during the growing season. $NH_4^+$ was extracted within 2 h of collection using 5 g of sieved (< 5 mm) soil in 20 mL of 2 M potassium chloride followed by colorimetric analysis (Astoria2 autoanalyzer, Astoria-Pacific International; Coweeta Hydrologic Laboratory, 2016). Soil pH was measured on 0–10 cm samples collected in three plots in winter of 2013 (each sample representing a composite of 20–25 2.5 cm diameter soil cores). Soil pH was determined directly after mixing 5 g soil with 10 mL 0.01 M calcium chloride (Coweeta Hydrologic Laboratory, 2016). Soil emission potential ($\Gamma_{soil}$) (unitless) was estimated directly from measured molar concentrations of $[H^+]$ and $[NH_4^+]$ as

$$\Gamma_{soil} = \frac{[NH_4^+]}{[H^+]}. \tag{3}$$

### 2.2.5 Above-canopy flux measurements

Above-canopy fluxes of $NH_3$ and $HNO_3$ were quantified from measurements of vertical concentration gradients conducted during the final (summer 2016) intensive when the tower was at maximum height and the greatest vertical separation of concentration measurements was achieved. Fluxes were determined using the modified Bowen ratio (MBR) method (Meyers et al., 1996) as

$$F = -K_c(z)\frac{dC}{dz}, \tag{4}$$

where $K_c$ and $dC/dz$ are the eddy diffusivity and vertical concentration gradient of the chemical species of interest. The value of $K_c$ for trace gases was assumed to be equivalent to the eddy diffusivity of heat ($K_t$), calculated as

$$K_c = K_t = -\overline{w't'}\frac{\Delta z}{\Delta t}, \tag{5}$$

where $\overline{w't'}$ is the kinematic surface heat flux measured by eddy covariance above the canopy (43.5 m a.g.l.), $\Delta t$ is the

air temperature difference between the levels at 43 and 35 m a.g.l., and $\Delta z$ is the height interval of air temperature measurements. Air temperature was measured using aspirated thermocouples, and $\Delta t$ was corrected for a small bias between the sensors determined by collocated comparison. Concentration gradients were determined from URG annular denuder measurements at 34.6 and 43 m a.g.l., as described above in Sect. 2.2.2. Given the complexity of the topography, no attempt was made to correct for potential roughness sublayer effects on either the eddy diffusivity or concentration gradients, which should be acknowledged as a source of uncertainty in the calculated $HNO_3$ and $NH_3$ fluxes. Additional detail on the gradient measurements is included in Sect. S3.

### 2.2.6 Seasonal and annual deposition budget

Speciated seasonal and annual total nitrogen deposition budgets were developed for the period August 2015 to August 2016. The wet-deposited components, including $NH_4^+$, $NO_3^-$, and total WSON, were directly measured. Speciated dry deposition was estimated by combining measured air concentrations, micrometeorology, biogeochemistry, and canopy physical characteristics within a box version of the Surface Tiled Aerosol and Gaseous Exchange (STAGE) model, which is an option in the Community Multiscale Air Quality Model (CMAQ) version 5.3 (Appel et al., 2021).

The STAGE model treats the deposition of gases and particles separately. The air–surface exchange of gases is parameterized as a gradient process and is used for both bidirectional exchange and dry deposition following the widely used resistance model of Nemitz et al. (2001) and Massad et al. (2010):

$$F = -f_{veg}\frac{\chi_a(z) - \chi_{z0}}{R_a} - \left(1 - f_{veg}\right)\frac{\chi_a(z) - \chi_g}{R_a + R_g}, \tag{6}$$

where $F$ is the net flux above the canopy (a negative value represents a net deposition flux and a positive value represents a net emission flux), which is the sum of the component cuticular ($F_{cut}$), stomatal ($F_s$), and ground ($F_g$) fluxes. The quantity $\chi_a(z)$ is the ambient concentration at a reference height ($z$), $\chi_{z0}$ is the concentration at height $d$ (displacement height) $+z_0$ (roughness length), $\chi_g$ is the ground layer compensation point, $R_a$ is the aerodynamic resistance between $z$ and $d + z_0$, $R_g$ is the total ground resistance including in-canopy aerodynamic resistance ($R_{inc}$), ground boundary layer resistance ($R_{bg}$), and soil resistance ($R_{soil}$) ($R_g = R_{inc} + R_{bg} + R_{soil}$), and $f_{veg}$ is the vegetation coverage fraction. The ground is fully covered by vegetation at our forested site, and $f_{veg}$ is therefore set to 1.

The quantity $\chi_{z0}$ is related to the canopy ($\chi_c$) and ground compensation points ($\chi_g$) according to

$$\chi_{z0} = \frac{\left(\frac{\chi_a(z)}{R_a} + \frac{\chi_c}{R_{bl}} + \frac{\chi_g}{R_g}\right)}{\left(\frac{1}{R_a} + \frac{1}{R_{bl}} + \frac{1}{R_g}\right)}, \tag{7}$$

where $R_{bl}$ is the leaf boundary layer resistance. $\chi_c$ follows Nemitz et al. (2001) but is modified to account for a cuticular compensation point ($\chi_{cut}$):

$$\chi_c = \frac{\begin{array}{c}\chi_a(z)(R_a R_{bl})^{-1} + \chi_s[(R_a R_s)^{-1} + (R_{bl} R_s)^{-1} \\ + (R_g R_s)^{-1}] + \chi_{cut}[(R_a R_{cut})^{-1} + (R_{bl} R_{cut})^{-1} \\ + (R_g R_{cut})^{-1}] + \chi_g(R_{bl} R_g)^{-1}\end{array}}{\begin{array}{c}(R_a R_{bl})^{-1} + (R_a R_s)^{-1} + (R_a R_{cut})^{-1} \\ + (R_{bl} R_g)^{-1} + (R_{bl} R_s)^{-1} + (R_{bl} R_{cut})^{-1} \\ + (R_g R_s)^{-1} + (R_g R_{cut})^{-1}\end{array}}, \tag{8}$$

where $\chi_s$ is the leaf stomatal compensation point, and $R_s$ and $R_{cut}$ are the stomatal and cuticular resistances, respectively.

The stomatal, cuticular, and ground compensation points ($\chi_s$, $\chi_{cut}$, $\chi_g$) are described according to Nemitz et al. (2000a) as a function of temperature ($T$) and the emission potentials ($\Gamma$):

$$\chi_{s,cut,g} = \frac{161\,512}{T} 10^{\frac{-4507.11}{T}} \Gamma_{s,cut,g}. \tag{9}$$

$\Gamma_{cut}$ is set to 0 in this study, and thus there is only deposition to leaf cuticles. For unidirectional exchange of gases other than NH$_3$, $\Gamma_s$ and $\Gamma_g$ are also set to 0. In the case of NH$_3$, the foliage and ground layers may act as a source or sink depending on the ratio of the ambient concentration to the respective compartment compensation point (Husted and Schjoerring, 1995). Here values of $\Gamma$ for NH$_3$ are derived from measurements of live vegetation, litter, and soil chemistry as described above. Values used in the base model simulation are described in Sect. 3.6, and the sensitivity of modeled NH$_3$ fluxes to $\Gamma$ is discussed in Sect. S5.

Formulas for each resistance component are summarized in Table S2. The resistances are largely estimated following Massad et al. (2010) with the following exceptions. The value of $R_s$ is based on the Noah (Chen and Dudhia, 2001) or P-X land surface scheme (Pleim and Xiu, 1995) in the Weather Research and Forecasting (WRF) model, and in this study, the P-X scheme is used. Deposition to wetted cuticular and ground surfaces considers the bulk accommodation coefficient, following Fahey et al. (2017), and can be a limiting factor for highly soluble compounds. The parameterization of $R_{inc}$ follows Shuttleworth and Wallace (1985) as do Massad et al. (2010) but here uses the canopy momentum attenuation parameterization from Yi (2008) and in-canopy eddy diffusivity following Harman and Finnigan (2007). This parameterization is similar to Bash et al. (2010), and detailed descriptions of model resistances can be found in the references mentioned above.

Dry deposition ($F$) of aerosol nitrogen (NH$_4^+$ and NO$_3^-$) is estimated as the product of the measured concentration ($C$) and the STAGE-modeled dry deposition velocity ($V_d$):

$$F = -V_d(z) \times C(z). \tag{10}$$

Aerosol dry deposition processes include gravitational settling, Brownian diffusion, surface impaction, and rebound.

Similar to gases, STAGE calculates the averaged $V_d$ for particles by summing the $V_d$ over vegetative or non-vegetative surfaces, weighted by vegetation cover fraction, which is 1 (full coverage) at Coweeta. $V_d$ for a particle with aerodynamic diameter $D_p$ is calculated as

$$V_d(D_p) = \frac{V_g}{1 - \exp\left[-V_g(R_a + R_{bp})\right]}, \tag{11}$$

where $R_{bp}$ is the boundary layer resistance for particles, and the gravitational settling velocity ($V_g$) is calculated as

$$V_g = \frac{\rho D_p^2 g}{18\mu} C_c, \tag{12}$$

where $\rho$ is the density of the aerosol, $g$ is the acceleration of gravity, $\mu$ is the air dynamic viscosity, and $C_c$ is the Cunningham slip correction factor. The turbulent transport processes are considered to be similar for gas and aerosol, and $R_a$ can be formulated based on the similarity theory relationships. Unlike deposition of gases, the boundary layer resistance usually dominates the aerosol deposition process as Brownian diffusion is much slower for particles than molecular diffusion is for gases (Pleim and Ran, 2011). Thus, $R_{bp}$ depends on the collection efficiency of the surface and can be determined following Shu et al. (2022) CE1 :

$$R_{bp} = \left[F_f u_*\left(Sc^{-\frac{2}{3}} + E_{im}\right)\right]^{-1}, \tag{13}$$

where $u_*$ is the friction velocity, and $Sc$ is the Schmidt number for particles. The quantity $E_{im}$ represents the collection efficiency by impaction and follows Slinn (1982) for vegetative canopies and Giorgi (1986) for smooth (non-vegetative) surfaces. The quantity $F_f$ is an empirical correction factor to account for increased deposition during convective conditions, parameterized as

$$F_f = V_{fac}\left(1 + 0.24\frac{w_*^2}{u_*^2}\right), \tag{14}$$

where $V_{fac}$ is an empirical constant representing the enhanced effects over vegetation canopies. For vegetative canopies, $V_{fac}$ is equal to the one-sided leaf area index (LAI) with a minimum value of 1, and for non-vegetative surface, a value of 1 is used. The quantity $w_*$ is the convective velocity scale (Deardorff velocity), defined as

$$w_* = \left(\frac{g}{T_v} z_i \overline{w't'}\right)^{\frac{1}{3}}, \tag{15}$$

where $T_v$ is virtual air temperature, $z_i$ is average depth of the mixed layer, and $\overline{w't'}$ is the measured kinematic surface heat flux.

A bulk $V_d$ for PM$_{2.5}$ is obtained by integrating size-resolved $V_d$ according to the particle size distribution. The

size distribution profiles for $NH_4^+$ and $NO_3^-$ are from measurements at eight Canadian rural forest sites (Zhang et al., 2008), and the size distribution for particulate organic nitrogen is estimated as an average of that for $NH_4^+$ and $NO_3^-$. Model sensitivities of particle nitrogen fluxes to assumed size distributions are discussed in Sect. S5.

The STAGE model is extracted from CMAQ v5.3 and executed in a one-dimensional mode. The prescribed surface parameters (e.g., $z_0$, $d$) were modified according to the site conditions. Continuous LAI data were extracted from the MODerate resolution Imaging Spectroradiometer (MODIS) global LAI product (MCD15A2H), which is generated daily at a 500 m spatial resolution, and each data point covers an 8 d period. The MODIS LAI (Fig. S6) was adjusted using in situ canopy measurements as described in Sect. S4. Hourly meteorological measurements, including air temperature, relative humidity, $u_*$, atmospheric pressure, precipitation rate, global radiation, and soil temperature and moisture, were used to drive STAGE. The Obukhov length, which is defined as

$$L = -\frac{u_*^3 T_v}{\left(kg\overline{w't'}\right)}, \tag{16}$$

where $k$ is the von Karman constant, was also calculated from micrometeorological measurements.

### 2.2.7 Air concentrations for dry deposition modeling

Air concentration data used for dry deposition modeling are summarized in Table 3. Hourly measurements of $HNO_3$ by DD_CL as well as $\Sigma AN$ and $\Sigma PN$ by TD-PC-CL were conducted for a full year and were therefore used directly for modeling. Over the 12-month sampling period, 18 %, 22 %, and 22 % of hourly $HNO_3$, $\Sigma AN$, and $\Sigma PN$ concentrations were missing or invalid, respectively. Missing data were replaced with the corresponding hour from the median diel profile comprised of days with $> 75$ % completeness. Surrogate formulas of nitrooxy-butanol ($C_4H_9NO_4$) and PAN ($C_2H_3NO_5$) were assumed for $\Sigma AN$ and $\Sigma PN$, respectively.

Continuous $NH_3$ concentrations were only measured during the last two intensives. Biweekly AMoN $NH_3$ measurements, with corrections for travel blanks, were used to establish a continuous 12-month time series of air concentration for annual deposition modeling. Ammonia concentrations are known to exhibit pronounced diel patterns, even in remote areas (Wentworth et al., 2016). Variability in air concentration interacts with diel cycles in surface wetness, turbulence, and other factors to influence diel patterns in air–surface exchange rates. To incorporate this interaction, the diel concentration pattern determined during spring and summer 2016 by MARGA $NH_3$ measurements (Fig. S7) was imposed on the biweekly AMoN $NH_3$ concentration. The hourly profile of $NH_3$ concentrations was normalized by the corresponding overall mean concentration to produce a normalized mean diel concentration profile. This profile was

then applied to each biweekly AMoN air concentration, temporally scaling the $NH_3$ concentration by time of day while maintaining the measured biweekly AMoN concentration. Gap filling of AMoN data was not required. Comparisons of $NH_3$ measurements are briefly discussed in Sect. S1, Fig. S2.

Hi-Vol measurements of speciated particulate N were only conducted during intensive periods to assess the relative contributions of inorganic and organic fractions to total water-soluble N. The CASTNET particulate $NH_4^+$ and $NO_3^-$ were used to provide a continuous 12-month time series of air concentration for annual deposition modeling. Concentrations of Hi-Vol and CASTNET measurements were shown to be comparable (Sect. S1, Fig. S4). For the annual time series, particulate organic nitrogen (PON) concentration was estimated based on speciated measurements during intensives, which showed that inorganic N accounts for $\sim 88$ % of WSTN on average. Weekly average PON concentration was estimated from the weekly CASTNET measurements assuming that $NH_4^+ + NO_3^-$ represents 88 % of total particulate nitrogen and PON represents the balance (12 %). Weekly concentrations were then expressed at the hourly timescale for modeling. Gap filling of CASTNET data was not required.

Components of the atmospheric reactive N budget that are not routinely measured at Coweeta and were not directly measured during SANDS include NO, $NO_2$, HONO, and $N_2O_5$. At the continental scale, regional model simulations suggest that NO, HONO, and $N_2O_5$ make minor contributions to the total dry deposition of reactive N ($\sim 2$ %), though the contribution of $NO_2$ is larger ($\sim 6$ %) (Walker et al., 2020). While NO, HONO, and $N_2O_5$ have been excluded from our modeling analysis, we have included an estimate of $NO_2$ concentration, CE2 from which dry deposition is estimated. The "other" fraction of $NO_y$ (i.e., $NO_y$–$HNO_3$–$\Sigma PN$–$\Sigma AN$) measured at Coweeta represents between 47 % (summer) and 76 % (winter) of total $NO_y$ on a seasonal basis. This "other" fraction includes NO, $NO_2$, HONO, $N_2O_5$, and some $NO_3^-$ but is likely dominated by $NO_2$. The measured diel profile of "other" $NO_y$ (Fig. S8) concentration shows the typical pattern indicative of morning and evening modes related to mobile $NO_x$ emissions. Winds at Coweeta are from the east-northeast during the morning, which is the direction of local residences, the town of Otto, NC, and US Highway 23. Winds are from the west-southwest during the evening, which is the direction of the Nantahala National Forest. Consistent with the diel profile of "other" $NO_y$, a much larger morning peak in $NO_2$ is therefore expected. To estimate the concentration of $NO_2$ from the measured "other" $NO_y$, we examined the ratio of $NO_2$ to the quantity $NO_y$–$HNO_3$–PANS–NTR (e.g., "other" $NO_y$) simulated by CMAQ (V5.2.1) for the Coweeta site over the year 2015; "PANS" represents total peroxy nitrates, and NTR represents other organic nitrates. Relative to the measured $NO_y$ species, PANS and NTR are assumed to represent $\Sigma PN$ and $\Sigma AN$, respectively. The ratio of CMAQ-estimated $NO_2$ to "other" $NO_y$ ranges from 0.51 during summer to 0.60 during winter.

**Table 3.** Summary of air concentration data sources for STAGE dry deposition modeling.

| Chemical species | Data source | Details |
|---|---|---|
| $NH_3$ | Measurement | AMoN measurement with diurnal profile imposed |
| $HNO_3$ | Measurement | Continuous DD-CL |
| $\Sigma PN$ | Measurement | Continuous TD-PC-CL. Assume molecular weight (MW = 121.1) of PAN ($C_2H_3NO_5$) |
| $\Sigma AN$ | Measurement | Continuous TD-PC-CL. Assume molecular weight (MW = 135.1) of nitrooxy-butanol ($C_4H_9NO_4$) |
| $NH_4^+$ | Measurement | CASTNET |
| $NO_3^-$ | Measurement | CASTNET |
| PON | Estimated based on measured $NH_4^+ + NO_3^+$ | Based on intensive direct measurements, assume PON represents 12 % of total $PON + NH_4^+ + NO_3^+$ |
| $NO_2$ | Estimated based on measured $NO_y$ | Based on ratio of $NO_2 / NO_y$ simulated by CMAQ V5.2.1 at Coweeta |

These seasonal factors were applied to the measured "other" $NO_y$ to estimate the hourly $NO_2$ concentration. Gap filling procedures for hourly "other" $NO_y$ follow those for $HNO_3$, $\Sigma PN$, and $\Sigma AN$ described above. Details of CMAQ V5.2.1 can be found in Table S3.

Regarding the use of measurements from different towers (Table 3) for inferential modeling, we acknowledge that differences in tower position on the landscape (i.e., within the forest –EFT – versus adjacent clearing – COW137) and the height of the measurement above the surface will be sources of variability in air concentrations. Given the complexity of the topography, no attempt was made to correct air concentrations for differences in measurement heights.

## 3 Results and discussion

### 3.1 Long-term trends in atmospheric N at Coweeta

Emissions of oxidized nitrogen ($NO_x$) and sulfur ($SO_x$) have declined significantly in the eastern US in response to the 1990 Clean Air Act Amendments. Trend data from the US EPA's National Emissions Inventory (NEI) indicate a nationwide decline of 74 % and 46 % for $SO_x$ and $NO_x$ emissions from the early 1990s to 2010s, respectively, comparing 1990–1994 to 2010–2014 annual averages (U.S. EPA, 2014). By contrast, $NH_3$ emissions have been reported as relatively unchanged or slightly increasing for the same periods (Ellis et al., 2013; Paulot and Jacob, 2014; Xing et al., 2013), depending on the location and region of the US. Declining $NO_x$ and $SO_x$ emissions resulted in decreasing trends in air concentrations of $HNO_3$ and $SO_2$ at Coweeta between the 1990s and 2010s (Fig. 2). Concentrations noticeably began to decline in 2008, the timeline of which likely indicates the effect of full implementation of the 2006 Tier 2 Gasoline Sulfur Program, as well as the enactment of the Clean Air Interstate Rule (CAIR), both of which aimed to further reduce $NO_x$ and $SO_x$ emissions (Sickles and Shadwick, 2015; LaCount et al., 2021). Compared to other species, $NH_3$ concentrations have only been measured at Coweeta for a relatively short period of time.

Atmospheric $NH_3$ reacts with acidic sulfate to form ammonium sulfate (($NH_4)_2SO_4$) or bisulfate (($NH_4)HSO_4$) aerosol. Under favorable thermodynamic conditions (low temperature and high RH), $NH_3$ in excess of acidic sulfate will react with $HNO_3$ to form ammonium nitrate aerosol ($NH_4NO_3$). Concentrations of $SO_4^{2-}$ at Coweeta have tracked $SO_2$, and subsequently $NH_4^+$ concentrations have declined substantially relative to early 1990s levels (Fig. 2). However, concentrations of $NO_3^-$ aerosol, which are relatively low at Coweeta, have not followed trends in $SO_4^{2-}$ and $NH_4^+$ (Fig. 2). Previous studies at other US sites have also reported non-proportional changes in $PM_{2.5}$ mass in response to $SO_2$ and $NO_x$ control strategies (Blanchard and Hidy, 2005; Sickles and Shadwick, 2015). Nonlinear reductions or increases in particulate $NO_3^-$ with coincident $SO_2$ and $NO_x$ emission reductions relate to the thermodynamic equilibrium of the $SO_4^{2-}$–$NO_3^-$–$NH_4^+$–$HNO_3$–$NH_3$ aerosol system. As ambient $SO_4^{2-}$ concentrations decline, the capacity for $NH_4^+$ formation (i.e., neutralization) also decreases, leaving additional $NH_3$ in the gas phase. Amounts of $NH_3$ in excess of acidic $SO_4^{2-}$ can subsequently react with $HNO_3$ to form $NH_4NO_3$, confounding the relationship between $NO_x$ emission reductions and atmospheric $NO_3^-$ concentrations.

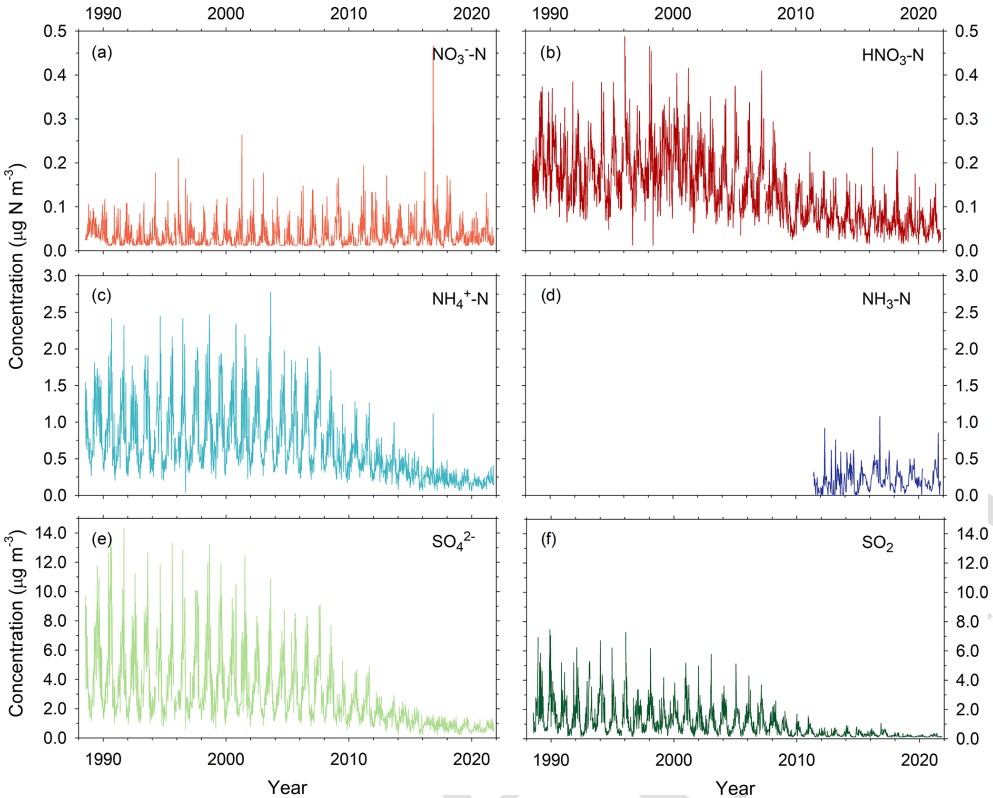

**Figure 2.** Long-term CASTNET (weekly, site COW137) and AMoN (biweekly, site NC25) air concentrations (as N) of $NO_3^-$ **(a)**, $HNO_3$ **(b)**, $NH_4^+$ **(c)**, $NH_3$ **(d)**, $SO_4^{2-}$ **(e)**, and $SO_2$ **(f)**.

The long-term trend in $NO_3^-$ wet deposition at Coweeta (Fig. 3) has tracked the downward trend in ambient $HNO_3$ concentration. Wet deposition of $NH_4^+$, however, shows no apparent trend, in contrast to the decline in $NH_4^+$ aerosol concentration. This pattern may relate to the combined effects of changes in regional $NH_3$ emissions, aerosol chemistry, and dry deposition rates on the long-term trend in atmospheric $NH_3$ concentrations. As noted above, declines in $SO_2$ emissions and $SO_4^{2-}$ aerosol result in less conversion of $NH_3$ to $NH_4^+$ aerosol, leaving more $NH_3$ in the gas phase. Furthermore, reduced air concentrations of acidic species such as $SO_2$ and $HNO_3$ result in lower dry deposition rates and subsequently less acidic deposition surfaces, which in turn reduces the deposition velocity (i.e., increases the atmospheric lifetime) of $NH_3$ (Sutton et al., 2003). In addition to changes in emissions, these two processes are thought to be at least partly responsible for the increases in $NH_3$ air concentrations that have been observed in some locations across the US (Butler et al., 2016; Yu et al., 2018; Yao and Zhang, 2019). While there is no discernable trend in $NH_3$ air concentrations over the relatively short period of record at Coweeta, a decline in wet deposition of $NH_4^+$ aerosol may have been offset to some extent by increased wet deposition of $NH_3$ gas (Asman, 1995), which is highly soluble, resulting in an overall lack of trend in $NH_4^+$ wet deposition at Coweeta over

time. Similar to other areas in the US (Li et al., 2016), the downward trend in $NO_3^-$ wet deposition has led to an increase in the relative contribution of reduced forms of N (i.e., $NH_x = NH_3 + NH_4^+$) to inorganic wet N deposition at Coweeta ($NH_4^+ : NO_3^-$, Fig. 3).

### 3.2 Wet deposition

Of the various forms of N in precipitation, ammonium was the most abundant inorganic species, contributing 47.0 % of WSTN in weekly samples ($N = 52$) on average, followed by $NO_3^-$ (41.7 %, Fig. 4, Table S4). The contribution of $NO_2^-$ was negligible. Organic compounds (WSON) contributed 11 % of WSTN on average, which is within the range of values (3 % to 33 %) reported for other locations in the US (Scudlark et al., 1998; Whitall and Paerl, 2001; Keene et al., 2002; Beem et al., 2010; Walker et al., 2012; Benedict et al., 2013). While concentrations of N compounds were generally higher during warm months, a seasonal pattern in the percent contribution of WSON to WSTN was not apparent. In a previous study at Coweeta (1994–1996), Knoepp et al. (2008) found that organic nitrogen contributed 21 % of total nitrogen in bulk (wet + dry) deposition samples. Differences between Knoepp et al. (2008) and SANDS results may be related to interannual variability or trends in rainfall composition over

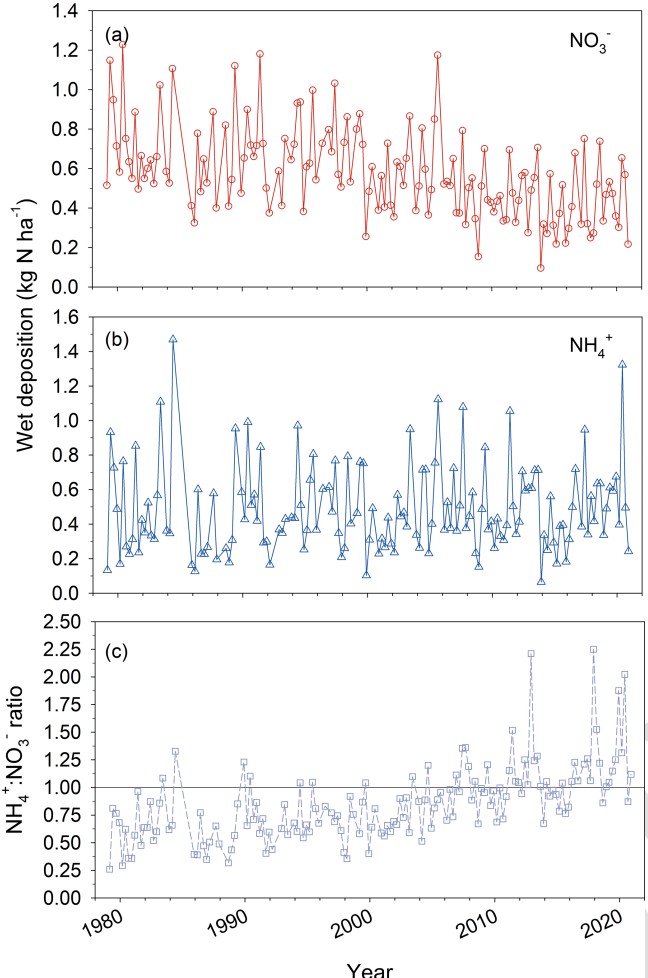

**Figure 3.** Long-term NTN NC25 measurements of seasonal $NH_4^+$ **(a)** and $NO_3^-$ **(b)** wet deposition along with the ratio of $NH_4^+$ to $NO_3^-$ as nitrogen and a 1 : 1 reference line **(c)**.

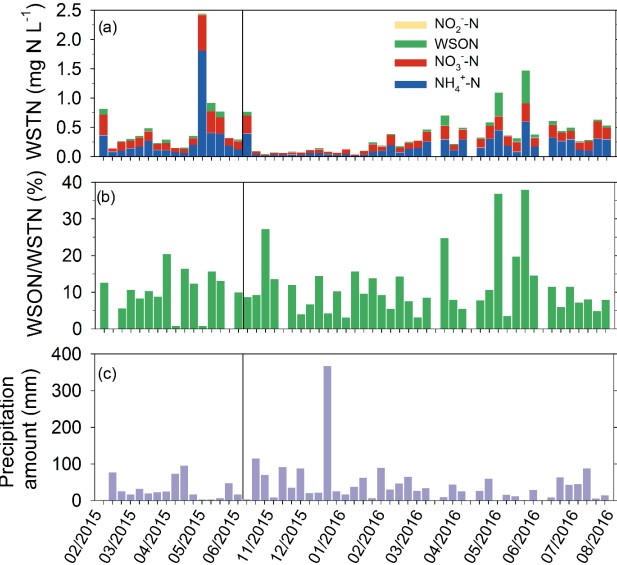

**Figure 4.** Concentrations of nitrogen species in weekly precipitation samples **(a)**, percentage contribution of WSON to WSTN in precipitation **(b)**, and precipitation amount **(c)**. The vertical line marks discontinuity due to missing data from 1 June to 19 October 2015.

the intervening 2 decades (e.g., Fig. 3), differences in collection method (wet only versus bulk deposition), or analytical techniques used for total N analysis (persulfate–UV digestion, Walker et al., 2012, versus total Kjeldahl N, Knoepp et al., 2008).

## 3.3 Air concentrations of oxidized N

The oxidized fraction of reactive nitrogen ($NO_y$) comprises a mixture of gaseous and particulate inorganic (NO, $NO_2$, $N_2O_5$, HONO, $HNO_3$, $NO_3^-$) as well as organic compounds. Owing to its large deposition velocity and typical atmospheric concentration, $HNO_3$ is the primary contributor to dry deposition of inorganic oxidized N (Walker et al., 2020). Much less is known about the dry deposition of oxidized organic nitrogen compounds (Walker et al., 2020). Peroxy nitrates (PNs) and alkyl and multifunctional nitrates (ANs) are formed during the photochemical oxidation of

volatile organic compounds (VOCs) in the presence of $NO_x$ ($NO_x = NO + NO_2$). While PNs exist in the gas phase, ANs can exist in the gas or particle phase and can be the dominant chemical sink for $NO_x$ in high-BVOC (biogenic VOC), low-$NO_x$ environments (Farmer and Cohen, 2008; Browne and Cohen, 2012; Paulot et al., 2012; Browne et al., 2013). Unlike PNs, ANs can also form at night via nitrate radical-induced oxidation of VOCs. Further, PNs and ANs have been shown to contribute significantly to the total $NO_y$ budget in geographically diverse rural and forested environments (e.g., Trainer et al., 1993; Nouaime et al., 1998; Farmer et al., 2008; Browne et al., 2013; Toma et al., 2019). Flux measurements at Blodgett Forest, CA, showed that PN dry deposition contributed 4 %–19 % of total N deposition at the site (Wolfe et al., 2009). Chemical transport modeling with current representation of the atmospheric oxidized nitrogen system suggests that PNs and ANs together contribute ∼ 6 % TS9 of total N deposition and ∼ 12 % TS10 of dry N deposition at the US continental scale compared to ∼ 21 % and 34 % for $HNO_3$ and ∼ 6 % and 9 % for particulate $NO_3^-$ (Walker et al., 2020).

The annual average concentration of $NO_y$ was 1.00 ppb (0.55 μg N m$^{-3}$), with the highest seasonal average concentration in the winter (1.32 ppb, 0.75 μg N m$^{-3}$) and lowest in the summer (0.64 ppb, 0.34 μg N m$^{-3}$) (Fig. 5, Table S5). The nearest rural $NO_y$ monitoring site is 85 km to the northwest at Look Rock in the Great Smoky Mountains National Park, where the annual concentration was 1.5 ppb over the same period (NPS, 2020). Similar to Coweeta, $NO_y$ concentrations at Look Rock are typically lowest during summer and highest in winter, though the seasonal cycle exhibits some

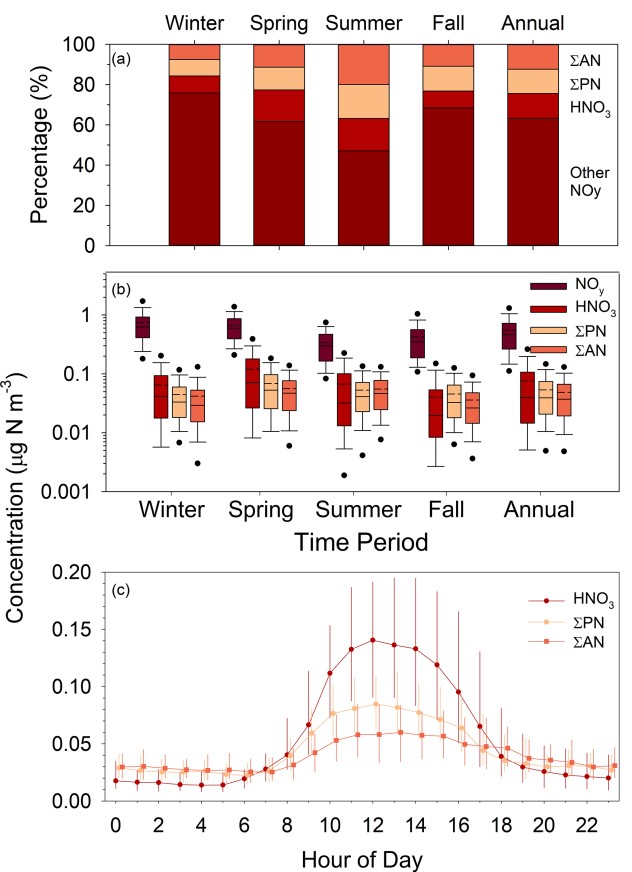

**Figure 5.** Seasonal and annual percent contribution of $HNO_3$, $\Sigma PN$, $\Sigma AN$, and other compounds to total $NO_y$ **(a)**; seasonal and annual box plots of $NO_y$, $HNO_3$, $\Sigma PN$, and $\Sigma AN$ – solid and dashed lines inside the box represent the median and mean, respectively; the top and bottom of the box represent 75th and 25th percentiles; whiskers represent 10th and 90th percentiles, and dots represent 5th and 95th percentiles. **(b)** Diurnal profiles of $HNO_3$, $\Sigma PN$, and $\Sigma AN$ – observations represent median hourly concentration, and bars represent interquartile range **(c)**. "Other $NO_y$" is calculated as $NO_y$–$HNO_3$–$\Sigma PN$–$\Sigma AN$, which, while primarily comprised of $NO_x$, includes $N_2O_5$, HONO, $NO_3^-$, and possibly other organics.

interannual variability. The annual mean concentrations of $HNO_3$, $\Sigma PN$, and $\Sigma AN$ determined by TD-PC-CL were 0.14 (0.08), 0.1 (0.06), and 0.09 (0.05) ppb ($\mu g\,N\,m^{-3}$), respectively (Fig. 5, Table S5). $HNO_3$ and $\Sigma PN$ concentrations peaked in spring, coincident with the seasonal peak in $O_3$ concentration, while concentrations of $\Sigma AN$ were similar in spring and summer. Diel patterns of $HNO_3$, $\Sigma PN$, and $\Sigma AN$ peaked during the day as expected for photochemical products. However, of the organic compounds, the ratio of peak daytime to minimum nighttime concentrations (Fig. 5) was much smaller for $\Sigma AN$ (2.3) compared to $\Sigma PN$ (3.9), possibly indicative of the additional nighttime formation of AN.

Annually, $HNO_3$ (12.8 %), $\Sigma PN$ (12.2 %), and $\Sigma AN$ (12.7 %) contributed approximately the same proportions of the atmospheric $NO_y$ load (Fig. 5, Table S6). Their collective contribution ($NO_z = HNO_3 + \Sigma PN + \Sigma AN$) to total $NO_y$ peaked during the summer (52.9 %) and reached a minimum during winter (24.2 %). The contributions of $\Sigma PN$ (16.7 %) and $\Sigma AN$ (20.0 %) exceeded $HNO_3$ (16.2 %) during summer when total $NO_y$ concentrations were lowest. Our results fall within the range of $NO_y$ budgets reported for other rural forested sites, in which $\Sigma PN$ and $\Sigma AN$ contribute $\sim 8$ %– 40 % (Nouaime et al., 1998; Farmer et al., 2008; Browne et al., 2013; Toma et al., 2019) and 10 %–22 % (Day et al., 2003; Farmer et al., 2008; Browne et al., 2013) of $NO_y$, respectively.

To put the SANDS period into context with longer-term variability of oxidized N concentrations at Coweeta, CAST-NET $HNO_3$ and $NO_3^-$ measurements for the period 2015– 2020 are summarized in Fig. 6 along with the SANDS period. We note here that $NH_4NO_3$ volatility on the CAST-NET Teflon filter can result in positive and negative biases in $HNO_3$ and $NO_3^-$, respectively, with larger biases expected under warmer conditions (Lavery et al., 2009). Studies have shown total $NO_3^-$ ($TNO_3 = HNO_3$ and $NO_3^-$) to be conserved, though some portion of the $NO_3^-$ collected by the CASTNET open-faced filter may be contributed by coarse particles. The partitioning of $TNO_3$ between gas and particulate phases is important, given the much larger deposition velocity of $HNO_3$ compared to $NO_3^-$. The CASTNET measurements reflect relatively low concentrations of both $HNO_3$ and $NO_3^-$, with $HNO_3$ exceeding $NO_3^-$ during all seasons. Particulate $NO_3^-$ concentrations are highest during cooler months, as expected, and negligible during the summer, a pattern that is consistent with observations from other networks across the southeast (Kim et al., 2015). Additionally, $TNO_3$ is primarily in the gas phase even during winter. Seasonal mean concentrations during the SANDS period fall within the interquartile range (IQR) of the 6-year period between 2015 and 2020, with SANDS annual and 6-year averages being very similar (Fig. 6). Seasonal and annual mean $HNO_3$ concentrations agreed closely with the CAST-NET measurements (Fig. 6 and Sect. S1).

### 3.4 Air concentrations of reduced N

Reduced forms of nitrogen ($NH_x$) represent another important component of the inorganic dry N deposition budget. At the continental scale, $NH_3$ dry deposition contributes $\sim 20$ % of total N deposition and $\sim 32$ % of dry N deposition, whereas the contributions from $NH_4^+$ aerosol are $\sim 4$ % and $\sim 6$ %, respectively (Walker et al., 2020). Similar to oxidized forms of N, the partitioning of mass between the gas ($NH_3$) and particulate ($NH_4^+$) phases affects the dry deposition rate of $NH_x$, given the larger deposition velocity of $NH_3$ relative to $NH_4^+$.

During 2015–2020, with 2020 being the most recent full year of AMoN data, average concentrations of $NH_3$ and $NH_4^+$ were similar (Fig. 7). Both species displayed a seasonal pat-

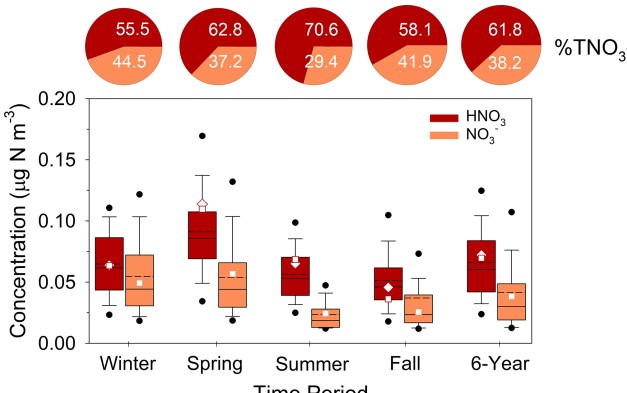

**Figure 6.** Summary of CASTNET $HNO_3$ and $NO_3^-$ concentrations (as N) from 2015–2020 during winter, spring, summer, and fall. Solid and dashed lines inside the box represent the median and mean, respectively. The top and bottom of the box represent 75th and 25th percentiles. Whiskers represent 10th and 90th percentiles, and dots represent 5th and 95th percentiles. "6-year" represents the statistics for the entire 6-year period. Squares and diamonds represent the seasonal and annual mean CASTNET ($HNO_3$ and $NO_3^-$) and continuous DD-CL $HNO_3$ for the August 2015–August 2016 modeling period, respectively. Pie charts represent the average percent contribution of $HNO_3$ and $NO_3^-$ to total $NO_3^-$ ($HNO_3 + NO_3^-$) expressed as nitrogen.

tern of lowest concentrations in the winter and higher concentrations during warm months. $NH_3$ concentrations peaked in summer, reflecting the temperature dependence of regional agricultural and biogenic emissions. $NH_4^+$ concentrations followed the seasonal cycle in $SO_4^{2-}$ concentrations, which were also similar in spring and summer and minimum in winter at Coweeta. The seasonal cycle of $NH_3$–$NH_4^+$ partitioning was driven more by $NH_3$ than $NH_4^+$, with the former exhibiting more seasonal variability. Hourly measurements conducted during spring and summer 2016 showed that $NH_3$ also displayed significant diel variability (Fig. S7), reaching a maximum around midday and a minimum overnight. Seasonal mean concentrations during the SANDS period fall within the IQR of the 6-year period between 2015 and 2020, with SANDS annual and 6-year averages being very similar. Concentrations of $NH_x$ were higher relative to $TNO_3$ during SANDS and over the longer term at Coweeta (Figs. 6 and 7).

### 3.5 Aerosol N composition

Ammonium was the most abundant inorganic species, contributing 86.8 % of WSTN ($N = 103$) on average (Fig. 8, Table S7). Organic compounds (WSON) contributed 11.5 % of WSTN, which is very similar to precipitation. The contributions of $NO_3^-$ and $NO_2^-$ were minor. Our study-wide average of percent WSON is slightly lower than measurements at other North American forest sites, including Duke Forest, North Carolina ($\sim$33 %, Lin et al., 2010), and Rocky

Mountain National Park (14 %–21 %) (Benedict, 2012), but is within the global range of 10 %–39 % (Cape et al., 2011). Similar to precipitation chemistry, there was no seasonal pattern in the percent contribution of WSON to WSTN in $PM_{2.5}$. Hi-Vol measurements of inorganic PM components compared well overall with collocated CASTNET measurements (Sect. S1, Fig. S4).

### 3.6 Biogeochemistry

Estimates of $NH_3$ emission potentials ($\Gamma$) for the ground and vegetation are needed to calculate compensation points ($\chi$) and fluxes in STAGE. Measurements of pH, $NH_4^+$, and corresponding $\Gamma$ of the leaves ($\Gamma_s$) and litter ($\Gamma_l$) are summarized in Fig. 9 and Tables S8 and S9. Measurements of $\Gamma_s$ are divided into green leaves collected during the growing season (spring and summer) and senescent leaves collected in October. $NH_3$ emission potentials ($\Gamma$) for green leaves ($\Gamma_s$) ranged from zero to 4070 with a median value (35.8) (Table S8) corresponding to a compensation point of $\chi_s = 0.25\ \mu g\ NH_3\ m^{-3}$ at 25 °C. Large intra-species variability of tissue pH and $NH_4^+$ was observed (Table S9), and separating by crown versus understory species did not reveal distinct differences between groups. Given the variability of the observations, the median $\Gamma_s$ was used for STAGE simulations. Senescence marks the translocation of N in leaves to storage tissues (Schneider et al., 1996). Along with a decline in photosynthetic activity, degradation of chlorophyll, and other metabolic changes, glutamine synthetase (GS) activity also declines (Pearson et al., 2002). Glutamine synthetase catalyzes assimilation of $NH_4^+$ into glutamine and is therefore important in regulating the pool of $NH_4^+$ available for exchange as $NH_3$ between the leaf and atmosphere as well as remobilizing organic N for storage during senescence. A decline in GS activity can thus result in increased leaf $NH_4^+$ concentrations (Pearson et al., 2002; Wang et al., 2011). Senescent leaves were similar to green leaves with respect to median tissue pH but showed higher concentrations of tissue $NH_4^+$. Median $\Gamma_s$ was correspondingly higher (113), equivalent to $\chi_s = 0.8\ \mu g\ NH_3\ m^{-3}$ at 25 °C. For STAGE modeling, the median $\Gamma$ for senescent leaves was used for $\Gamma_s$ during the fall.

Regard our method for estimating $\Gamma_s$, the fundamental assumption is that the chemistry of the bulk leaf tissue is representative of the apoplast. While a number of studies have shown positive correlations between bulk tissue chemistry, apoplastic chemistry, and independently quantified compensation points (David et al., 2009; Hill et al., 2002; Massad et al., 2010; Mattsson and Schjoerring, 2002; Mattsson et al., 2009), absolute differences between $\Gamma_s$ derived from bulk tissue versus apoplast measurements can be large. For example, Sutton et al. (2009) and Personne et al. (2015) both show that ratios derived from bulk tissue chemistry exceed those derived from apoplast chemistry. We did not perform experiments to validate the use of bulk tissue chemistry as a proxy

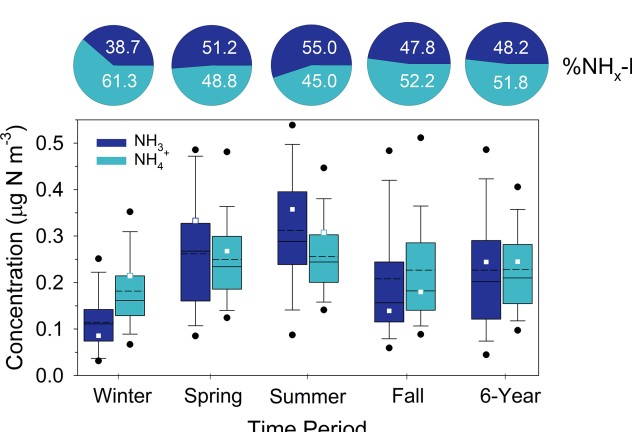

**Figure 7.** Summary of AMoN $NH_3$ and CASTNET $NH_4^+$ concentrations (as N) from 2015–2020 during winter, spring, summer, and fall. Solid and dashed lines inside the box represent the median and mean, respectively. The top and bottom of the box represent 75th and 25th percentiles. Whiskers represent 10th and 90th percentiles, and dots represent 5th and 95th percentiles. "6-year" represents the statistics for the entire 6-year period. Squares represent the seasonal and annual mean concentration for the August 2015–August 2016 modeling period. Pie charts represent the average percent contribution of $NH_3$ and $NH_4^+$ to total $NH_x$ ($NH_3 + NH_4^+$) expressed as nitrogen. AMoN concentrations were adjusted by subtracting the mean travel blank for the 6-year period.

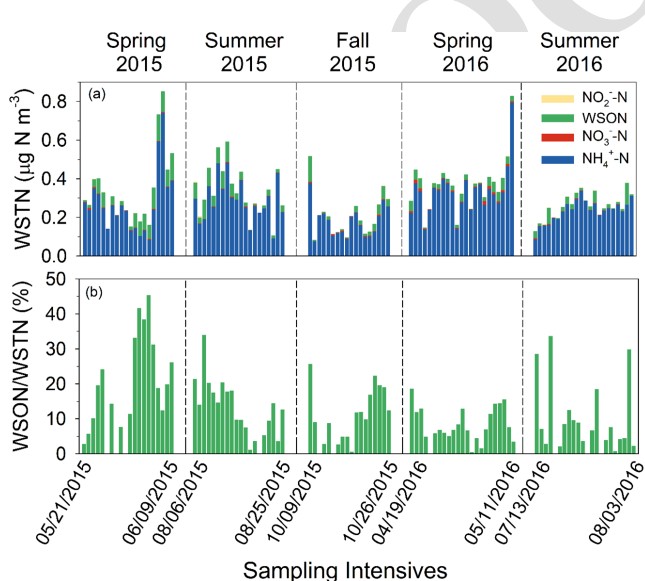

**Figure 8.** Contributions of N aerosol species to WSTN in 24 h Hi-Vol $PM_{2.5}$ samples during seasonal SANDS intensives **(a)** along with the percentage contribution of WSON to WSTN **(b)**.

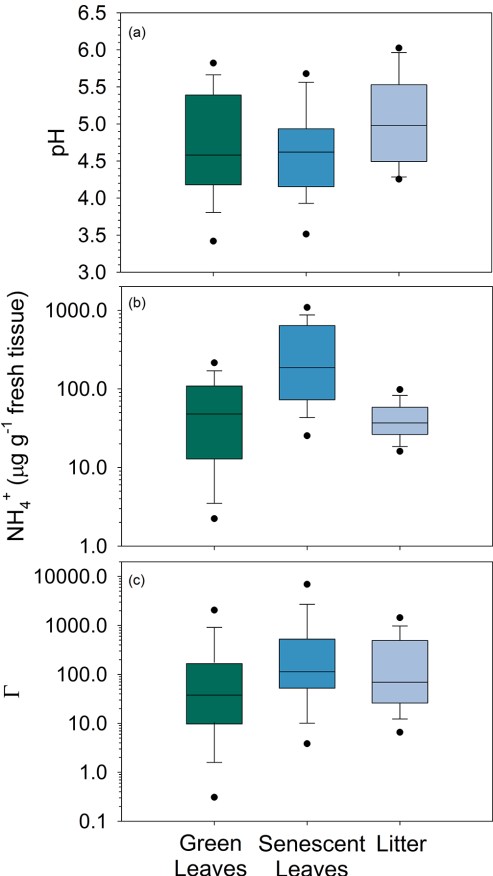

**Figure 9.** Box plots of pH **(a)**, $NH_4^+$ **(b)** concentration ($\mu g\,g^{-1}$ fresh tissue), and equivalent emission potential ($\Gamma$) **(c)** in tissue of green leaves, senescent leaves, and litter on the forest floor. The solid line inside the box represents the median. The top and bottom of the box represent 75th and 25th percentiles. Whiskers represent 10th and 90th percentiles, and dots represent 5th and 95th percentiles.

for apoplast chemistry and acknowledge this source of uncertainty. However, our estimates of $\Gamma_s$ appear reasonable in the context of the range of existing observations (cited above and summarized by Massad et al., 2010) and the general relationship between $[NH_4^+]_{bulk}$ and $\Gamma_s$ put forth by Massad et al. (2010, Eq. 6). Measurements on bulk tissue are less labor-intensive and therefore more tempting than measurements on the apoplast. However, more studies comparing the two procedures are needed to extend the meta-analysis of Massad et al. (2010) to a wider range of natural ecosystems, particularly deciduous forests.

Leaf litter on the soil surface has been shown to be a source of $NH_3$ to the atmosphere in both natural and agricultural ecosystems (Nemitz et al., 2000b; David et al., 2009; Hansen et al., 2013). As litter decomposes, mineralization of organic N is a source of $NH_4^+$, some of which is lost to the overlying air as $NH_3$. Litter $NH_4^+$ concentrations were similar to green leaves but lower than senescent leaves (Fig. 9). However, the pH was higher than both green and senes-

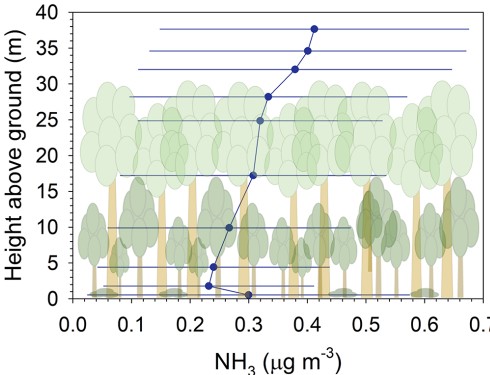

**Figure 10.** Vertical concentration profiles of $NH_3$. The mean (filled circle) and standard deviation (bars) of concentrations are shown for $N = 76$ daytime profiles.

cent leaves. The resulting median $\Gamma_1$ (69.3) was larger than green leaves but smaller than senescent leaves, equivalent to $\chi_1 = 0.49\,\mu g\,NH_3\,m^{-3}$ at $25\,°C$. Litter $\Gamma$ was much larger than that of the underlying soil. Average (0–10 cm soil depth)
soil pH (4.18) and $NH_4^+$ (1.21 mg N kg$^{-1}$) correspond to $\Gamma_{soil} \sim 10$ at a soil mass wetness of $0.1\,g\,g^{-1}$, equivalent to $\chi_{soil} \sim 0.1\,\mu g\,NH_3\,m^{-3}$ at $25\,°C$. This very low $\Gamma_{soil}$ results from the low pH of the shallow soil.

Vertical profiles of air concentrations within and above
10 the canopy were measured to investigate patterns of air–surface exchange with specific ecosystem compartments (i.e., canopy crown, understory, and ground). A detailed analysis of bidirectional N fluxes is forthcoming; thus, we limit the discussion of these data to $NH_3$ in the context of interpret-
15 ing patterns observed in the biogeochemical emission potentials and their prescription in the STAGE model. Nitric acid, $NH_4^+$, and $NO_3^-$ showed expected decreasing concentrations from above the canopy to the forest floor, indicative of deposition. While $NH_3$ profiles showed patterns of deposition to
20 the crown and understory, concentrations near the forest floor indicated both emissions and deposition (Fig. 10). Of the 76 daytime profiles measured, 40 % showed decreases down to the forest floor, and 60 % showed an increasing concentration from approximately the lower understory ($\sim 5$ m a.g.) to the
25 forest floor. The former pattern is interpreted as deposition to the forest floor, and the latter is interpreted as emission. Thus, the mean profile suggests a source of $NH_3$ at the ground. The very low $\Gamma_{soil}$ suggests that emission from the soil is unlikely given such a low pH. The leaf litter layer, which in-
30 dicates a much higher emission potential ($\Gamma_1$) than the soil, is a more likely source of $NH_3$. This hypothesis is consistent with Hansen et al. (2013, 2017), in which emissions of $NH_3$ from a beech (*Fagus sylvatica*) forest after leaf fall were attributed to the decomposition of new litter. Similar to our
site, the underlying soil also had low pH (4–5). Given our observations, we used $\Gamma_1$ (median $= 69.3$, Table S8) rather than $\Gamma_{soil}$ as the ground emission potential ($\Gamma_g$) in STAGE.

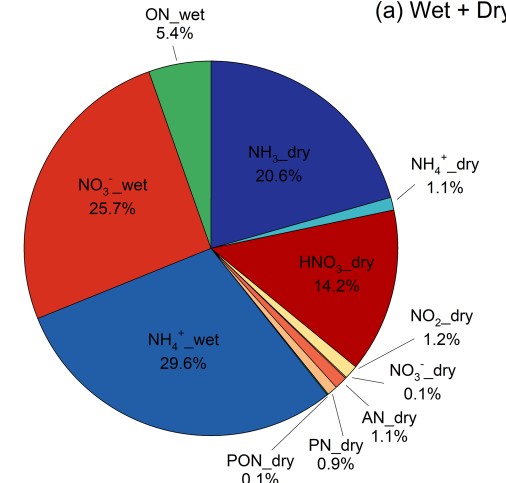

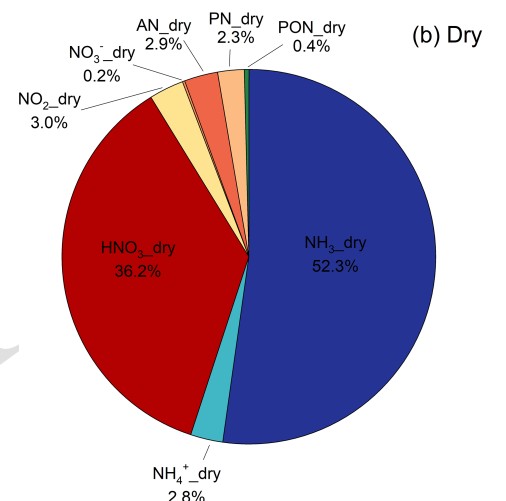

**Figure 11.** Speciated annual total (wet and dry) **(a)** and dry **(b)** deposition showing the percent contribution of individual components. TS11

### 3.7 N deposition budget

Total annual N deposition for the period August 2015–August 2016 was 6.7 kg N ha$^{-1}$ (Fig. 11). Over this period,
wet deposition contributed 60.7 % of total N deposition, of which $NH_4^+$ was the primary component (29.6 %). Wet deposition of organic N contributed 5.4 % of the total N deposition budget. Dry deposition accounted for 39.3 % of total deposition, of which $NH_3$ was the primary contributor (20.6 %).
Reduced forms of inorganic N were the largest contributor to the budget (51.2 %), with oxidized inorganic and organic N contributing 41.2 % and 7.6 % of total N deposition, respectively. Dry deposition of organic N made a small contribution (2.2 %) to the total deposition budget.
Ammonia is the most important contributor to the dry deposition budget (52.3 %) and differs from the other species

in that it is exchanged bidirectionally between the ground, canopy, and atmosphere. TS12

Seasonal net canopy-scale and component fluxes are shown in Fig. S9. The mean net flux ($F_{net}$) is downward (i.e., deposition) during all seasons, generally following the seasonal pattern of the atmospheric $NH_3$ concentration. The cuticular flux ($F_{cut}$), which is unidirectional in STAGE, is the dominant deposition pathway and ranged from $-97.7$ ng N m$^{-2}$ s$^{-1}$ (deposition) to near zero. The contribution of $F_{cut}$ to the total net flux ranged from 84.7 % in spring to $\sim 100$ % during fall. The stomatal flux ($F_s$) is bidirectional, ranging from $-4.5$ ng N m$^{-2}$ s$^{-1}$ (deposition) to 2.3 ng N m$^{-2}$ s$^{-1}$ (emission), with the largest fluxes occurring during warmer periods of the growing season when the stomatal resistance is lowest. The stomatal flux is smaller than $F_{cut}$ for several reasons. First, $R_s$ is generally larger than $R_{cut}$. Also, in the current model formulation the cuticular compensation is $X_{cut} = 0$. Thus, the $NH_3$ concentration gradient between air above the leaf ($X_{leaf}$) and $X_s$ is smaller than for $X_{cut}$. Finally, low LAI and large $R_s$ in winter and fall as well as offsetting bidirectional fluxes in spring and summer result in a relatively small mean stomatal deposition flux ($F_s$) across seasons. $F_g$ is also bidirectional, ranging from $-3.9$ ng N m$^{-2}$ s$^{-1}$ (deposition) to 2.5 ng N m$^{-2}$ s$^{-1}$ (emission). Fluxes are largest during spring as atmospheric $NH_3$ begins to increase with warmer temperatures but before peak LAI is reached, after which the denser canopy increases the in-canopy aerodynamic ($R_{inc}$) and air-side ground boundary layer resistances ($R_{bg}$) (Table S2), thereby decreasing $F_g$. On an annual scale, $F_g$ and $F_s$ make similar contributions to ($\sim 3.0$ %) $F_{net}$.

Nitric acid was the second-largest component of dry deposition, contributing 36.2 % of the total. While $HNO_3$ deposits more rapidly than $NH_3$ (Table S10), the overall importance to the dry N budget is constrained by relatively low air concentrations at this remote forest site. Particulate species made much smaller contributions to the budget due to much lower deposition velocities ($V_d$ = flux–air concentration) relative to their gaseous counterparts (Table S10). For example, while $NH_4^+$ contributed more to the atmospheric $NH_x$ load than $NH_3$, (Fig. 7), the $NH_x$ flux budget was regulated by the much more rapid exchange of $NH_3$ between the forest and atmosphere relative to $NH_4^+$. A similar example was observed for oxidized N. While $NO_2$ represents an important fraction of the oxidized N concentration budget via its contribution to "other $NO_y$", $NO_2$ deposits much less rapidly than $HNO_3$ (Table S10), thereby contributing a relatively small fraction (3.0 %) of the dry N flux. Of the organic N species, AN contributed slightly more (2.9 %) to dry N deposition than PN (2.3 %) owing to a higher deposition velocity (Table S10). Similar to particulate $NH_4^+$ and $NO_3^-$, PON made a small contribution to dry N deposition (0.4 %) due to its low $V_d$ (Table S10). Reduced forms of N accounted for the majority of dry N deposition (55.1 %), with oxidized inorganic and

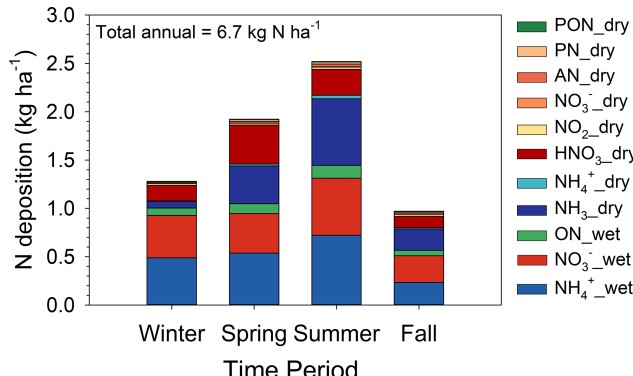

**Figure 12.** Seasonal speciated deposition budget. $N_r$ species are listed in the legend as defined in the text, along with indication of the deposition pathway (dry or wet).

organic forms of N contributing 39.4 % and 5.5 %, respectively. TS13

Total N deposition peaked during the summer (2.5 kg N ha$^{-1}$) and reached a minimum in the fall (1.0 kg N ha$^{-1}$) (Fig. 12). Wet deposition exceeded dry deposition during all seasons. Seasonal variability in wet deposition was primarily driven by precipitation amount, whereas dry deposition was influenced by seasonality in air concentrations of the primary $N_r$ species (Figs. 6 and 7), LAI (Fig. S6), turbulence, and other surface characteristics. Ammonia fluxes followed the seasonal pattern of air concentration, peaking in the summer and reaching a minimum in winter. Concentrations and fluxes of $HNO_3$ peaked in the spring and reached a minimum in the fall. Deposition velocities, which can be thought of as the concentration-normalized flux, peaked during the summer and reached a minimum during winter for most N species. This pattern likely reflects the combined effect of seasonal cycles in LAI and turbulence characteristics. The seasonal pattern of $V_d$ for $HNO_3$ differed slightly from the other species, peaking in spring and reaching a minimum in fall. In contrast to other N species, $HNO_3$ deposition is limited by turbulent transfer, with the canopy (surface) resistance assumed to be zero. The pattern of $HNO_3$ $V_d$ thus partially reflects seasonal patterns in wind speed and the degree of turbulent mixing above the canopy.

## 3.8 Evaluation of the dry deposition model

While total uncertainty in the dry deposition budget cannot be rigorously quantified (Walker et al., 2019a), the sensitivity of the model to parameterizations and key inputs can elucidate important aspects of model uncertainty and inform a potential range of dry deposition estimates. Here we undertake such an exercise by evaluating several alternative modeling scenarios to assess the sensitivity of fluxes and total dry deposition to assumptions regarding LAI, $NH_3$ emission potentials ($\Gamma_{s,1}$), $NH_3$ cuticular resistance ($R_{cut,dry}$), and par-

ticle size distribution. We focus on $NH_3$, as it is the most important component of the dry deposition budget and more complex with regard to air–surface exchange processes than the other species. Sensitivity tests are summarized in Sect. S5 and Table S11. Of the scenarios tested, increasing $\Gamma_l$ and $\Gamma_s$ within the range of observations and reducing $R_{cut,dry}$ within the variability reported by Massad et al. (2010) exerted the largest control over the dry deposition flux, establishing a range of total dry deposition from 2.0 (increasing $\Gamma_{s,l}$) to 3.1 (decreasing $R_{cut,dry}$) kg N ha$^{-1}$ around the base value of 2.6 kg N ha$^{-1}$. The corresponding percent contribution of $NH_3$ to total dry N deposition ranges from 36.6 % to 58.5 % (base 52 %), and the contribution of dry to total wet + dry deposition ranges from 33.0 % to 43.0 % (base 39.4 %). Our results point to the need for a better understanding of the processes of cuticular exchange and the importance of adequately characterizing the magnitude and variability of vegetation and litter emission potentials in forests.

Another method of evaluating model behavior is comparison with measured $V_d$. During the final summer intensive, a small dataset ($N = 19$ observations) of $V_d$ was determined from daytime measurements of vertical concentration gradients above the canopy using the MBR method. Measured $V_d$ was compared to $V_d$ derived from the STAGE model for overlapping periods and the maximum possible $V_{dmax}$ as $1/(R_a + R_b)$. Of the 19 MBR measurements, four $NH_3$ profiles exhibited emissions (6.8 to 22.4 ng $NH_3$ m$^2$ s$^{-1}$), which were not reproduced by STAGE. Analysis of the meteorological conditions during the MBR measurements suggests that emissions tend to occur during the warmest periods with lowest relative humidity. This would correspond to periods when $R_{cut}$ and $X_s$ are high and may indicate that the model is underestimating $F_s$ emissions during these periods. Excluding the four emission periods, $V_d$ estimated from MBR and STAGE agree reasonably well (Fig. S11). As is the case for STAGE, resistance-based models typically assume that $HNO_3$ deposits at the rate of $V_{dmax}$ (i.e., $R_c = 0$). As shown in Fig. S11, fluxes measured during summer 2016 showed MBR $V_d$ for $HNO_3$ larger than $NH_3$, as expected, but lower than $V_{dmax}$. This apparent nonzero $R_c$ could result from a real nonzero $R_c$ caused, for example, by equilibrium of $HNO_3$ and $NO_3^-$ on leaf surfaces (Nemitz et al., 2004a). This pattern may also reflect the influence of flux divergence related to $NH_4NO_3$ evaporation in the canopy crown, which would reduce the magnitude of the downward vertical gradients, and therefore the measured $V_d$, of $HNO_3$ and $NH_3$ (Nemitz et al., 2004b). In this analysis, concentrations of $NO_3^-$ (mean 0.08 µg m$^{-3}$) were much lower than $HNO_3$ (mean 0.47 µg m$^{-3}$), and $NO_3^-$ gradients were therefore difficult to resolve, precluding a definitive explanation of $HNO_3$ $V_d < V_{dmax}$. Ignoring potentially significant uncertainties related to the measurement of chemical and temperature gradients within the roughness sublayer, our results suggest that periods of $NH_3$ emission during the day, particularly at higher air temperature and lower humidity, may be underestimated. Our results also reinforce the need for temporally extensive measurements of concentrations and fluxes of $HNO_3$, $NH_3$, and $NO_3^-$ to examine exchange processes and uncertainties related to chemical flux divergence.

## 3.9 Spatial and temporal representativeness of the deposition budget

The complexity of atmospheric flows in mountainous terrain influences the spatial variability of wet and dry deposition processes (Lehner and Rotach, 2018). As the deposition budget presented above is based on measurements from the lowest elevation portion of the Coweeta Basin, the degree to which the budget is spatially representative must be considered. Potential effects on dry deposition were assessed by characterizing the magnitude and spatial variability of $HNO_3$ and $NH_3$ concentrations along an elevation gradient (Fig. 1, Table 1) from the lower to upper portions of the Coweeta Basin. It should be noted that $HNO_3$ concentrations at NC25 were measured by CASTNET, while $HNO_3$ passive samplers were used at the other locations. Concentrations are summarized in Fig. 13, in which the sites are ordered left to right from lowest to highest elevation. Nitric acid and $NH_3$ concentrations increase slightly with elevation, an explanation for which is not obvious. Nitric acid concentrations are highest at Screwdriver Knob, which is distinct from the other sites in that the measurement tower was situated on a relatively narrow exposed ridge. The measurements are therefore higher above the surrounding vegetation than at the other sites. Overall, variability of air concentrations across sites, even including SK, is sufficiently small such that spatial variability of dry deposition across the basin would likely be driven more by variability in meteorology than air concentrations.

A quantitative assessment of the effects of airflow on dry deposition across the basin is not possible, but the work of Hicks (2008) illustrates the relevant effects in the context of the resistance analogy for $V_d$. Over flat homogeneous terrain, flux to vegetation is driven by turbulent diffusion in the vertical direction above the canopy and horizontal advection is assumed to be zero. In the extreme case of airflow approaching a steep forested slope, horizontal flow penetrates the canopy and the transfer of material (deposition) to the canopy elements becomes dominated by horizontal advection and filtration rather than vertical diffusion. In the context of $V_d$, this situation is analogous to the aerodynamic resistance ($R_a$) approaching zero. Taking $HNO_3$ as an example under the typical assumption that the canopy resistance is $R_c = 0$, $V_d$ becomes limited by the quasi-laminar boundary layer resistance at the vegetation surfaces ($R_b$). Following the analysis of Hicks (2008), $V_d$ for $HNO_3$ could be enhanced by a factor of $[1 + (R_a/R_b)]^{1/2}$ (Hicks, 2008). Using median values of $R_a$ and $R_b$ from our modeling period, this would increase $V_d$ for $HNO_3$ by a factor of $\sim 1.4$. For gases that have a significant $R_c$, enhancements will be smaller. Topographical relief

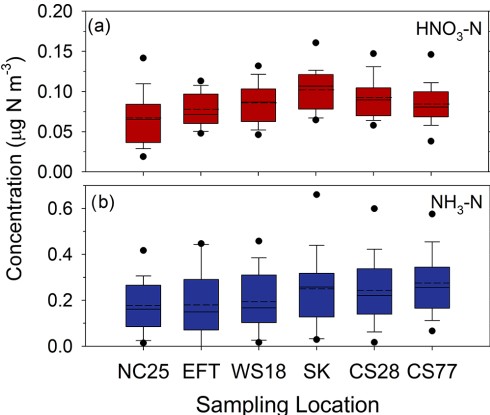

**Figure 13.** Concentrations (as N) of $HNO_3$ **(a)** and $NH_3$ **(b)** measured at different elevations, increasing from left to right (see Fig. 1 and Table 1), across the Coweeta Basin. Solid and dashed lines inside the box represent the median and mean, respectively. The top and bottom of the box represent 75th and 25th percentiles. Whiskers represent 90th and 10th percentiles, and dots represent 95th and 5th percentiles.

across the Coweeta Basin may be gentle enough such that the flow separation described in the previous example is limited to certain areas and meteorological scenarios. However, as Hicks (2008) points out, flow complexity in mountainous areas has the overall effect of increasing $V_d$, with areal weighted fluxes being highly dependent on the topographical characteristics specific to the study area. Other effects related to katabatic flows (Novick et al., 2016) and diel patterns of hillside shading that drive temperature-related processes such as $NH_3$ compensation points introduce additional uncertainties.

The results of Knoepp et al. (2008) show that spatial patterns of wet deposition across the Coweeta Basin follow patterns of precipitation amount, which increase with elevation. In their study, bulk deposition of $NH_4^+$, $NO_3^-$, and total organic nitrogen was measured from 1994–1996 at sites ranging in elevation from 788 to 1389 m. Annual precipitation depth and bulk deposition increased by 25 % from the lowest to the highest elevation. This increase in precipitation with elevation is consistent with the 75-year analysis of Coweeta climatological data by Laseter et al. (2012), which showed that the annual precipitation amount at 1398 m was 32 % greater than at 686 m. In our study, wet deposition was measured at the NC25 site at 686 m and therefore represents a lower wet deposition rate than would occur in higher-elevation portions of the basin. An approximate 35 % enhancement in both wet and dry deposition for the highest elevations within the basin would correspond to a total N deposition rate of $9.0\,kg\,N\,ha^{-1}\,yr^{-1}$ based on our estimate of $6.7\,kg\,N\,ha^{-1}\,yr^{-1}$ for the lower basin.

Regarding the temporal representativeness of the deposition budget calculated here, wet deposition of inorganic N ($NO_3^- + NH_4^+$) during our 12-month model period ($3.69\,kg\,N\,ha^{-1}$) agrees well with the mean annual deposition rate measured at NTN site NC25 ($3.72\,kg\,N\,ha^{-1}$) over the period 2015–2020, with 2020 being the most recent full year of observations reported at the time of this analysis. Air concentrations of $NO_3^-$ and $HNO_3$ (Fig. 6) as well as $NH_3$ and $NH_4^+$ (Fig. 7) during our model period are also similar to the 6-year (2015–2020) mean concentrations measured by CASTNET and AMoN. In this context, our results are deemed temporally representative of the most recently available complete years of monitoring data.

## 4 Conclusions

Due to the success of the Clean Air Act, air concentrations and wet deposition of reactive N at Coweeta are the lowest observed since the beginning of routine monitoring in the late 1970s. However, even at historically low levels, our results show that $N_r$ deposition remains highly ecologically relevant in the context of critical loads. Our estimate of total $N_r$ deposition of $6.7\,kg\,N\,ha^{-1}\,yr^{-1}$ is near the upper-end estimate of mass-balance-derived critical loads ($2.8$ to $7\,kg\,N\,ha^{-1}\,yr^{-1}$) recently reported for spruce–fir, beech, and mixed deciduous forests by Pardo et al. (2018) in the nearby Great Smoky Mountains National Park. Our result also falls within the range of empirical critical loads of N for combined tree health and biogeochemical responses ($3–8\,kg\,N\,ha^{-1}\,yr^{-1}$) as well as changes in mycorrhizal fungi spore abundance, community structure, and community composition ($5–12\,kg\,N\,ha^{-1}\,yr^{-1}$) in eastern temperate forests (Pardo et al., 2011).

A key feature of the deposition budget derived for Coweeta is the predominance of reduced forms ($NH_x$) of inorganic nitrogen (51.2 %) over oxidized inorganic N (41.2 %). TS14 Reductions in deposition of $NH_x$ will be needed to achieve the lower-end estimates of critical N loads ($\sim 3\,kg\,N\,ha^{-1}\,yr^1$) for southern Appalachian forests. This presents a challenge, as emissions and air concentrations of $NH_3$ remain unregulated. Our results also show that organic forms of N make a nontrivial contribution (7.6 %) to total N deposition, primarily via wet deposition. It is noted, however, that the gas-phase dry component of deposition only considers oxidized forms as alkyl and peroxy nitrates, excluding contributions from reduced (i.e., NH) organic compounds. While our results represent an advancement in accounting for organic dry $N_r$ deposition in total $N_r$ deposition, the application of new measurement technologies (Walker et al., 2019b) for broader chemical speciation of organic forms of dry $N_r$ deposition is needed.

Our results underscore the need for long-term measurements of reactive chemical fluxes, as well as the coupling of atmospheric and biogeochemical measurements, to improve air–surface exchange models. Novel measurements that more directly elucidate the role of cuticular exchange of

$NH_3$ and more temporally extensive measurements of leaf $NH_3$ emission potentials are particularly needed. For forest ecosystems, a physically representative parameterization for resistance to $NH_3$ diffusion through the leaf litter layer and more temporally extensive measurements of the litter $NH_3$ emission potential combined with more thorough understanding of litter decay dynamics are also needed. For sensitive ecosystems located in mountainous and other topographically complex landscapes, which includes much of the Class I wilderness area in the US, identification of locations suitable for micrometeorological flux measurements will be challenging. Novel flux measurement methods and application of in situ models, including translation of measurements from more ideal to complex locations, will likely be needed. Furthermore, long-term flux datasets are needed to assess the interactive effects of changing air quality and climate on both atmosphere–biosphere exchange and ecosystem response to deposition (e.g., Van Houtven et al., 2019).

*Code availability.* Code is available at https://doi.org/10.5281/zenodo.7667344 (Bash and Wu, 2023).

*Data availability.* NADP, AMoN, and CASTNET data are available at the websites referenced herein (see Sect. 2.1 for more information TS15). Other data appearing in figures and tables are available at https://catalog.data.gov/dataset/epa-sciencehub (EPA ScienceHub, 2023).

*Supplement.* The supplement related to this article is available online at: https://doi.org/10.5194/bg-20-1-2023-supplement.

*Author contributions.* JTW: conceptualization, formal analysis, methodology, funding acquisition, project administration, validation, visualization, writing. XC: formal analysis, investigation, methodology, validation, writing. ZW: formal analysis, investigation, methodology, software validation, writing. DS: investigation, formal analysis. RD: investigation, formal analysis, validation. AD: data curation, investigation, methodology, resources. ACO: conceptualization, formal analysis, methodology, validation. EE: data curation, funding acquisition, formal analysis, methodology, validation, resources. JB: formal analysis, methodology, software. JK: data curation, investigation. MP: conceptualization, funding acquisition, resources. JI: formal analysis, investigation, writing. CFM: conceptualization, funding acquisition, resources, writing.

*Competing interests.* The contact author has declared that none of the authors has any competing interests.

*Disclaimer.* The views expressed in this article are those of the authors and do not necessarily represent the views or policies of the U.S. EPA. The findings and conclusions in this publication are those of the authors and should not be construed to represent any official USDA or U.S. Government determination or policy.

*Publisher's note:* Copernicus Publications remains neutral with regard to jurisdictional claims in published maps and institutional affiliations.

*Acknowledgements.* We gratefully acknowledge field and laboratory support from USDA Forest Service staff at the Coweeta Hydrologic Laboratory, including Christine Sobek, Patsy Clinton, Chuck Marshall, and Cindi Brown. David Kirchgessner (retired, U.S. EPA) tirelessly supported field and laboratory activities during SANDS intensives. Lee Nanny (former U.S. EPA) and Mark Barnes (U.S. EPA) supported field intensives and logistics. We also appreciate the support of Kevin Mishoe (Wood, Inc.) and Christopher Rogers (Wood, Inc.) for support of CASTNET field activities and data management, respectively.

*Financial support.* This research has been supported by the U.S. Environmental Protection Agency (Intramural) and the USDA Forest Service, Southern Research Station, Coweeta Hydrologic Lab.

*Review statement.* This paper was edited by Ivonne Trebs and reviewed by Chris Flechard and one anonymous referee.

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

**Remarks from the language copy-editor**

CE1   The requested correction has been adapted to our house standards. Please confirm that your intended meaning is intact.

CE2   The comma is grammatically correct in this context.

**Remarks from the typesetter**

TS1   Please note that the changes in values have not been inserted as they should be approved by the editor to ensure a transparent review process. Please provide a short explanation about these corrections that can be forwarded by us. Thank you very much in advance for your help.

TS2   Please see previous remark regarding editor approval.

TS3   Please see previous remark regarding editor approval.

TS4   Please see previous remark regarding editor approval.

TS5   Please see previous remark regarding editor approval.

TS6   Please confirm dates throughout the paper. If you meant 11 January 2022, please let me know.

TS7   Please see previous remark regarding editor approval.

TS8   Please see previous remark regarding editor approval.

TS9   Please see previous comment regarding editor approval.

TS10   Please see previous comment regarding editor approval.

TS11   Please see previous remark regarding editor approval for the new figure file.

TS12   Please see previous remark regarding editor approval for all values changed in these two paragraphs.

TS13   Please see previous remark regarding editor approval for all values changed in this paragraph.

TS14   Please see previous remark regarding editor approval.

TS15   Please confirm.

TS16   Please confirm.

TS17   Please confirm.