# Peer review of "Atmospheric Deposition of Reactive Nitrogen to a Deciduous 2 Forest in the Southern Appalachian Mountains"

_Biogeosciences, 2022_

## Author Comment (AC1)

**Reviewer 1 comments and response**

The manuscript by Walker et al. presents results from a study investigating atmospheric
deposition of reactive nitrogen to a deciduous forest at the USDA Forest Service Coweeta
Hydrologic Laboratory in the southern Appalachian Mountains. The authors use several well-
established measurement methods to differentiate between oxidized and reduced as well as
organic and inorganic compounds found in wet and dry deposition. Finally, they apply a bi-
directional resistance-based model driven with the observed measurements of Nr air
concentrations, micrometeorology, canopy structure, and biogeochemical parameters to present
the full reactive nitrogen budget for the site.

While the character of the paper is a report-style compilation of results from a multitude of
methods rather than following a clear scientific question, the authors do a great job in thoroughly
describing the complexity of reactive nitrogen field investigations and long-term observation.
Though continuous eddy-covariance observations are not included, the study represents the state-
of-the-art in Nr monitoring and data interpretation. I particularly appreciate the inclusion of field
investigations of the ammonia emission potential of green and senescent leaves as well as from
litter, which is crucial for model parameterization and rarely conducted. The results are put into a
broader context and discussed with regard to air quality regulations in the past, e.g. reduction in
oxidized N is now clearly visible. Method uncertainties are sufficiently considered and
presented.

The text is very well written and easy to follow. Figures are clear and easy to grasp. The
supplemental material is useful and the selection of graphs and tables that were put into this
section is good. This is the most comprehensive single-site study I am aware of and definitely
deserves publication.

I only have a few, rather minor, points that should be considered before final presentation in the
BG journal:

**Response:**  We sincerely thank the reviewer for their comments and questions.  We have
addressed each in detail below.

**Comment:** With regard to Section 2.2.7, how exactly were the NH3 data from hourly
measurements used to impose diurnal variability on the biweekly data to be used as hourly input
for the model? It is stated in line 393 that continuous NH3 concentrations were only measured
during the last two intensives (in spring and summer, I guess?). The diurnal variability is known
to be driven by temperature, humidity, light availability, phenology, etc., how was the amplitude
of the variability from these two campaigns transferred to the other – probably much cooler –
seasons?

**Response:**  The reviewer is correct that continuous hourly measurements were only conducted
during the last two intensives.  To account for diurnal variability in the $NH_3$ air concentration,
the diel concentration pattern determined during the spring and summer intensives
(Supplemental Figure S7) was imposed on the bi-weekly AMoN $NH_3$ concentration. First, the
hourly profile of $NH_3$ concentrations was normalized by the corresponding overall mean concentration to produce a normalized mean diel concentration profile. This profile was then
applied to each biweekly AMoN air concentration, temporally scaling the $NH_3$ concentration by
time of day while maintaining the measured biweekly AMoN concentration.  In this way, the
hourly time series derived for the entire year from AMoN measurements displays the diel
variability observed during the spring and summer.

The reviewer is correct, the amplitude of the diel concentration profile would be expected to
change throughout the year in response to a number of factors, e.g., temperature, local $NH_3$
emissions, boundary layer dynamics, biogeochemistry.  Our approach does not incorporate the
seasonality that may be observed in the diel profile for winter and fall.  However, while the
amplitude of the diel cycle may differ from our observations during these seasons, the
seasonality of the air concentration on a longer averaging period (i.e., two weeks) is reflected in
the AMoN concentrations.  An alternative approach that could be employed in the future would
be to derive diel profiles from a chemical transport model at shorter than seasonal time scales,
perhaps monthly, to scale the AMoN measurement to the hourly time scale for flux modeling.

**Comment:** The method section is very informative, but quite long. I'm wondering whether it
would make sense to put all detailed descriptions from 2.2.1 up to 2.2.5 into the supplement, just
adding a few sentences to 2.2 what has been done and referring to the respective part in the
Supplemental Material. It's not a must, but would significantly reduce length and better highlight
the findings given the potential readership of people who work in conservation and are likely
more interested in the results and their interpretation than in every technical detail of the
methodology.

**Response:**  We appreciate the reviewer's comment and admittedly it was difficult to decide how
much detail on methods to include in the primary text as opposed to supplemental. Ultimately,
we felt that the methods themselves, and particularly the combination thereof to assess the
deposition budget, were worth describing in the main text along with the results.

Other:

**Comment:** Introduction: I suggest adding information on measurement period, length, etc.

**Response:** This information will be added.

**Comment:** Line 41-42: "many areas" and "some regions", please specify where, e.g. near
hotspots of animal husbandry, chemical industry, etc.

**Response:** Additional detail will be added.

**Comment:** Line 153: Is 8 m the correct height? What was the reason for this height?

**Response:**  Yes, 8 m is correct.  This was the height of a permanent tower immediately adjacent
to the shelter housing the TD-PC-CL instrument.  The tower was used opportunistically and 8 m
happened to be the maximum height.  Reviewer 2 also asked about the choice of measurement
heights and corresponding treatment of measurements at different heights for flux modeling.  As discussed in the response to reviewer 2, we made no correction for differing measurement
heights and will clarify that point and limitation in the revised text.

**Comment:** Line 170: What was the selection criteria for the two respective heights?

**Response:** In this case, the heights were chosen to maximize the separation between sampling
boxes to maximize the concentration gradient. 34 m is just above the top of the canopy and 37.5
m and 43.5 m were the total height of the tower in 2015 and 2016, respectively.  Ultimately, the
MARGA data were not used for gradient flux calculations because colocation experiments to
remove systematic bias between the sample boxes were not successful.  As described in the
primary text, gradient fluxes were instead calculating using measurements from the time-
integrated denuder measurements.

**Comment:** Line 178: "to the analytical box for analysis Ion Chromatography (IC)", is there a
word missing?

**Response:** Yes, this will be corrected.

**Comment:** Line 270: Check for consistency in unit notation: "$g^{-1}$ tissue" vs. "kg tissue$^{-1}$"

**Response:**  Thank you for pointing out this inconsistency.  We will make the correction here and
check throughout.

**Comment:** Line 312: Delete "is $R_a$" after "$z_0$"

**Response:** OK

**Comment:** Line 449: Is RH defined before?

**Response:** RH will be defined at line 236 in the revised text.

**Comment:** Line 505-506: Do two decimal places reflect the measurement accuracy?

**Response:**  Two decimal places are appropriate for the corresponding detection limits (0.018 –
0.038 ppb).

**Comment:** Line 605-606: 61.4% wet plus 38.7% dry deposition equals 100.1%, check rounding.

**Response:** Thank you for point this out.  Rounding will be checked here and throughout.

**Comment:** Line 617: Can a bit more explanation given why stomatal fluxes are so low
compared to cuticular fluxes?

**Response:**  We will add more detail to the explanation of the low stomatal fluxes compared to
the cuticular flux.  First, the stomatal resistance is generally larger than the cuticular resistance
even during the summer when the stomatal resistance reaches a minimum. Second, the gradients that drive the leaf-level stomatal ($F_s$) and cuticular ($F_{cut}$) fluxes are defined as $X_s-X_l$ and $X_{cut}-X_l$,
respectively. $X_s$ and $X_{cut}$ are the stomatal and cuticular compensation points, respectively, and $X_l$
is the concentration of $NH_3$ above the leaf.  In the current model formulation $X_{cut}$ is zero and $X_s$
is non-zero and a function of the stomatal emission potential and temperature.  Thus, the
concentration gradient is always larger for the cuticular versus the stomatal pathway.  Together
the larger stomatal resistance and smaller concentration gradient ($X_s-X_l$) result in $F_s < F_{cut}$.

**Comment:** Line 708: Why would the aerodynamic resistance become zero at steep forested
slopes? $R_a$ is turbulence and wind speed driven, so why would it approach zero?

**Response:** Over flat homogeneous terrain, vertical exchange between the atmosphere and
vegetation is driven by vertical turbulent diffusion.  In the traditional resistance analogy, the
resistance to turbulent transfer is referred to as the aerodynamic resistance ($R_a$).  Hicks (2008)
describes an extreme example in which horizontal flow approaching a steep slope penetrates the
canopy.  In this example, the transfer of material (deposition) to the canopy elements becomes
dominated by horizontal advection and filtration rather than vertical diffusion.  Thus, this
situation can be described as analogous to $R_a$ tending to zero, i.e., no aerodynamic resistance to transfer.

**References**

Hicks, B.B., 2008. On estimating dry deposition rates in complex terrain. Journal of Applied
Meteorology  and Climatology, 47, 1651 – 1658.

---

## Author Comment (AC2)

Response to reviewer 2

Reviewer's comments on Biogeosciences manuscript "Atmospheric Deposition of Reactive
Nitrogen to a Deciduous Forest in the Southern Appalachian Mountains" by J.T. Walker

General Comments

This manuscript describes the atmospheric reactive nitrogen (Nr) deposition budget over a
deciduous forest in the Southern Appalachian Mountains. Extensive measurements of the wet
and dry deposition components of total deposition of inorganic and organic, reduced and
oxidized, gas- and aerosol-phase Nr, are reported for the years 2015-2016, when intensive
measurement campaigns were conducted at a forest site in Coweeta Basin as part of the SANDS
programme.

Wet deposition was measured in straightforward manner by precipitation collectors, while dry
deposition was mostly modelled from measured air concentrations and surface-atmosphere
exchange (inferential) modelling. Some aerodynamic gradient-flux measurements were made for
gases and aerosols over a limited period of time, providing measured reference points to assess
the performance of the surface-atmosphere exchange model.

The detailed, speciated, multi-season, multi-site measurements of most of the dominant and also
less documented (e.g. organic) forms of Nr concentrations in air and water offer a rare,
measurement-based glimpse into the diversity of all Nr forms contributing to total Nr deposition
over a US forest, and into the technical challenges and solutions implemented to close the
deposition budget.

The data from the 2015-2016 SANDS intensive campaigns are examined in the light of multi-
year or multi-decadal observation datasets from CASTNET, AMoN, NADP and EPA
measurement networks, showing the decreases observed in total Nr deposition to the site over the
last 3-4 decades (mostly from a long-term reduction in NOx emissions), but highlighting the
increasing importance of reduced nitrogen in total deposition and the continued exceedance of
critical loads for this ecosystem. The paper is therefore very well suited for the readership and
scope of Biogeosciences.

The manuscript presents a very detailed and clear description of the measurement methods used
in the extensive data collection, and assimilation by inferential modelling, which I find very
useful for this type of paper, where the objective and scope include a thorough methodological
component to document the manifold aspects required to compute a comprehensive Nr
deposition budget. Such methodological aspects deserve not to be trivialized and glossed over,
and will be useful to other researchers in this field, confronted by the complexities of total Nr
deposition budgetting.

The paper is very well written, and I have only very few and minor comments before
recommending eventual publication in Biogeosciences.

**Response:** We sincerely thank the reviewer for their comments and questions.  We have
addressed each in detail below.

Specific Comments

**Comment:** line 153: some gas and aerosol components of total Nr were measured at 1-10m
above ground , while the canopy height is 30m. I presume this means the samplers were located
in a clearing of the forest. How was this accounted for in inferential modelling of dry deposition,
knowing that the model supposes that concentrations are measured above the canopy, and that
concentrations measured in a (small) clearing are likely to represent sub-canopy levels rather
than above-canopy concentrations? Was there a correction scheme to account for this effect?

**Response:** This is a good point. As the reviewer points out, some measurements were taken
above the canopy on the eddy flux tower while another set of measurements was collected in an
open area nearby the tower. We did not make any attempt to correct for potential differences in
concentration due to measurement height but will clarify this point in the text, noting potential
dilution effects of sub-canopy drainage into the open area, particularly at night.

**Comment:** line 265 and lines 564-569: the Gamma_s parameter in the bi-directional NH3
exchange model should represent the emission potential (NH4+/H+) of the apoplast, i.e. the
inter-cellular fluid that is exposed to the air within sub-stomatal cavities. Here the assumption is
made (implicitly) that the NH4+/H+ ratio of bulk tissue extracts (whole leaf, i.e. whole cells inc.
vacuole, symplast and apoplast all mixed) is equal to the apoplastic emission potential. Many
publications have previously reported vastly different NH4+/H+ ratios for bulk tissue and
apoplast (e.g. Sutton et al, Biogeosciences, 6, 2907–2934, 2009, fig.7 over grassland, 1-2 orders
of magnitude difference; Wang et al., Plant Soil (2011) 343:51–66, conclude p64: "...bulk leaf
tissue Đ“ can not be used as a tool to predict the potential NH3 exchange of beech leaves" ).
Some publications do assert that there is a positive relationship between bulk and apoplastic
Gamma ratios, and bulk ratios are of course much more easily measured than apoplastic
extraction methods, so it is tempting to use the bulk tissue ratio as a proxy, for simplicity. Do the
authors have evidence that it is justified in the case of this particular forest ecosystem? They do
present a sensitivity analysis later on, using upper and lower percentiles, but I didn't see any
explicit discussion of why or how the bulk tissue ratio could be used as a proxy for the apoplastic
ratio. Please comment.

**Response:** The reviewer raises an important point here. We are indeed using the $NH_4^+/H^+$ ratio
(stomatal emission potential, $\Gamma_s$) from measurements on leaf bulk tissue as a proxy for that of the
apoplast. As rightly pointed out by the reviewer, while a number of studies have shown positive
correlations between bulk tissue chemistry, apoplastic chemistry, and independently quantified
compensation points (David et al., 2009; Hill et al. 2002; Massad et al. 2010; Mattsson and
Schjoerring 2002; Mattsson et al. 2009), absolute differences between ratios derived from bulk
tissue versus apoplast measurements can be large. For example, Sutton et al. (2009) and
Personne et al. (2015) both show that ratios derived from bulk tissue chemistry exceed those
derived from apoplast chemistry. As will be clarified in the text, we did not perform experiments
to validate the use of bulk tissue as a proxy for apoplast chemistry.

To put our bulk tissue derived $\Gamma_s$ into broader context, our results fall within the range, but on the
lower end, of $\Gamma_s$ reported for forests in the meta-analysis of Massad et al., 2010. Using data from studies in which $\Gamma_s$ was reported along with the concentration of $NH_4^+$ in bulk tissue, Massad et
al. (2010) derived a general relationship:

$\Gamma_s = 19.3 \times \exp{(0.0506 \times [NH_4^+]_{bulk}}$                                             (1)

where $[NH_4^+]_{bulk}$ is the concentration of $NH_4^+$ in leaf tissue ($\mu g\ g^{-1}$ tissue).  Using our measured
median value of $[NH_4^+]_{bulk}$ in equation (1) gives $\Gamma_s = 210$, which is larger than our tissue derived
median value of $\Gamma_s = 36$ but on the same order as the 75[th] percentile ($\Gamma_s = 171$) used as the upper
value in our model sensitivity analysis.  In general, our estimates of $\Gamma_s$ are reasonable in the
context of existing observations and the general relationship between $[NH_4^+]_{bulk}$ and $\Gamma_s$ put forth
by Massad et al. (2010).  That being said, we certainly acknowledge the reviewer's point
regarding uncertainty in the validity of our use of bulk tissue chemistry as a proxy for apoplastic
chemistry and will expand on this point in the text.  As the reviewer points out, measurements on
bulk tissue are easier and therefore more tempting to use compared to apoplastic extractions.
More studies comparing apoplast and bulk tissue derived $\Gamma_s$ are needed to extend the meta-
analysis of Massad et al. (2010) to a wider range of natural ecosystems, particularly deciduous
forests.  This point will also be emphasized in the revised text.

**Comment:** line 647: "This pattern largely reflects the seasonal cycle in leaf area index". Could
seasonal patterns in wind speed, turbulence, surface wetness (rainfall), also contribute to
seasonal Vd patterns, aside from LAI?

**Response:**  Yes, we agree that seasonal patterns in other drivers could also contribute to
seasonality in $V_d$ and will clarify this point in the text.

line 758-9: "more temporally extensive measurements of the litter $NH_3$ emission potential are
also needed". I would add that a better understanding (and modelling) of the leaf litter decay
dynamics, constrained by weather (temperature, moisture) are needed if one aims to reproduce
litter N emissions in surface exchange models.

**Response:**  Thank you for the comment.  We agree and will add this point to the text.

Technical corrections

**Comment:** line 290: add "by eddy covariance" after "heat flux measured..."

**Response:**  OK

**Comment:** lines 427-428: the sentence " To estimate the concentration of NO2 from the
measured "other" NOy, we examined the ratio of NO2 to the quantity NOy – HNO3 – PANS –
NTR (e.g., "other" NOy) simulated by CMAQ (V5.2.1) for the Coweeta site over the year
2015418-419..." feels a little like a repeat of lines 418-419

**Response:** Thank you for point this out.  We will shorten the sentence at line 418 to eliminate
redundancy.

**Comment:** line 442, figure 2 and figure S9: the decrease of SOx emissions and concentrations over 30 years had a large impact on NHx chemistry, and is useful to explain the NHx trends. It would be good to show the SO2/SO4= data of Fig S9 in Fig.2 of the main text, alongside long-term trends of Nr?

**Response:** Good suggestion. We will add the sulfur time series to Figure 2 of the main text.

**Comment:** line 505, fig. 5: NOy concentrations are expressed in ppb, it might be good to harmonize with the rest of the figures as µg m-3 (easier to compare NOy with TNO3- and NHx of figs 6-7, for example) ?

**Response:** Agreed. Concentrations will be harmonized to $\mu g\ m^{-3}$ in the revised text.

**Comment:** line 517: suggest change "the same proportions of the NOy budget..." to "the same proportions of the atmospheric NOy load ..." ? The word budget may suggest deposition ?

**Response:** Agreed. Wording will be changed to "atmospheric NOy load"

**Comment:** line 631, similar to above, suggest change to "NH4+ contributed more to the atmospheric NHx load than NH3..."

**Response:** Agreed. Wording will be changed to "atmospheric NHx load"

**Comment:** line 556: "The contributions of NO3 - and NO2- were negligible." This refers to Fig. 8, but in the top part (a) of Fig. 8, I don't see that NO3- was negligible (here, WSON is negligible, as is NO2-). And subsequently, "Organic compounds (WSON) contributed 11.6% of WSTN...", again that is not what the top figure shows, but it is what the lower part (b) of Fig. 8 apparently shows. There is a contradiction between the two parts (a) and (b): which is WSON, and which is NO3- ? Amend text if neccessary.

**Response:** This was a mistake in the color coding of the top chart and will be corrected.

**Comment:** Fig. 8 caption: suggest change to "Contributions of N aerosol species to WSTN..."

**Response:** Thank you. Wording will be changed as suggested.

**References**

David M, Loubet B, Cellier P, Mattsson M, Schjoerring JK, Nemitz E, Roche R, Riedo M, Sutton MA (2009) Ammonia sources and sinks in an intensively managed grassland canopy. Biogeosciences 6: 1903–1915. doi:10.5194/bg-6-1903-2009

Hill PW, Raven JA, Sutton MA (2002) Leaf age-related differences in apoplastic NH4+ concentration, pH and the NH3compensation point for a wild perennial. J Exp Bot 53: 277–286. doi:10.1093/jexbot/53.367.277

Massad RS, Nemitz E, Sutton MA (2010) Review and parameterization of bi-directional
ammonia exchange between vegetation and the atmosphere. Atmos Chem Phys 10:10359–
10386. doi:10.5194/ acp-10-10359-2010

Mattsson M, Schjoerring JK (2002) Dynamic and steady-state responsesof inorganic nitrogen
pools and NH3 exchange in leaves of Lolium perenne and Bromus erectus to changes in root
nitrogen supply. Plant Physiol 128:742–750. doi:10.1104/pp.010602

Mattsson M, Herrmann B, Jones S, Neftel A, Sutton MA, Schjoerring JK (2009) Contribution of
different grass species to plant-atmosphere ammonia exchange in intensively managed grassland.
Biogeosciences 6:59–66. doi:10.5194/bg-6-59-2009

Personne, E., Tardy, F., Genermont, S., et al. (2015) Investigating sources and sinks for ammonia
exchanges between the atmosphere and a wheat canopy following slurry application with trailing
hose. Agricultural and Forest Meteorology 207:11-23.

Sutton, M. A., Nemitz, E., Milford, C., Campbell, C., Erisman, J. W., Hensen, A., Cellier, P.,
David, M., Loubet, B., Personne, E., Schjoerring, J. K., Mattsson, M., Dorsey, J. R., Gallagher,
M. W., Horvath, L., Weidinger, T., Meszaros, R., Dämmgen, U., Neftel, A., Herrmann, B.,
Lehman, B. E., Flechard, C., and Burkhardt, J. (2009) Dynamics of ammonia exchange with cut
grassland: synthesis of results and conclusions of the GRAMINAE Integrated Experiment,
Biogeosciences6: 2907–2934. https://doi.org/10.5194/bg-6-2907-2009.